# Comparative analysis of zebrafish fear responses to eight different fish species using three-dimensional locomotion-tracking assays

Kevin Adi Kurnia[1,2], Gilbert Audira[1,2], Michael Edbert Suryanto[1,2], Tzong-Rong Ger[3,4,*] and Chung-Der Hsiao[1,2,4,5,*]

## ABSTRACT

Zebrafish (*Danio rerio*) are widely used in neurobehavioral research due to their translational relevance in studying fear. Eight different fish species and variations were tested to induce fear responses in zebrafish, including one positive control (convict cichlid, *Amatitlania nigrofasciata*) and negative control (tiger barb, *Puntigrus tetrazona*) through a shared-environment test. The observation was done in three dimensions (3D) and two dimensions (2D) to assess the impact of dimensionality on the outcome. A single-camera system was used to capture two viewpoints by mirror reflection installed above the fish tank and reconstructed to 3D using F3LA software. Zebrafish showed a similar behavioral response towards Demason's cichlid (*Pseudotrophus demasoni*) and threadfin acara (*Acarichthys heckelii*) as they did to *A. nigrofasciata*, with some minor differences, and a lesser response to green *Gymnocorymbus ternetzi*, during the shared-environment tests. Meanwhile, presence of *B. melanopterus* caused zebrafish to have a higher tendency to freeze and display higher entropy, similar to an anxiety-like response. We found no correlation between behavioral response and the body size of the test fishes. However, a correlation was observed when we tested convict cichlids of different ages. Finally, zebrafish color preference was also observed through the use of *G. ternetzi* with different body colors as test fish, with the zebrafish preferring orange and red *G. ternetzi* and mostly avoiding green *G. ternetzi*. We found use of 3D observation superior to 2D observation because several important endpoints are obtainable only from certain viewpoints.

KEY WORDS: Fear, Behavior changes, Zebrafish, Social interaction, 3D locomotion

## INTRODUCTION

Fear is a response to the presence of threat that can be observed in humans and animals and is related to survival (Lang et al., 2000; Daniel-Watanabe and Fletcher, 2022). It is identified as a response to an immediate threat and can be acquired in a multitude of ways, with the most common being from experiencing the cause (i.e. being bitten by a dog) or seeing others experiencing it. Fear responses are unique to each individual, and may include perspiration and changes in respiration as well as more obvious responses such as freezing, fleeing, or avoiding the source of threat (Delgado et al., 2006). Scientists use animal models to study fear responses by observing behavioral changes in certain conditions (Gerlai, 2020).

Zebrafish are an aquatic animal model often used due to their capability as a high-throughput behavioral model, in addition to low cost, ease of maintenance (Lieschke and Currie, 2007), and availability of detailed genomics (Patowary et al., 2013). Zebrafish are also known to be able to exhibit anxiety-like responses, stress responses, and fear responses, observable through their behavioral phenotypes and hormone expressions (Egan et al., 2009; Jesuthasan, 2012). Zebrafish behavioral phenotypes have been described in detail by Kalueff et al., 2013. Behaviors such as erratic movement, increased shoal cohesion, diving, and freezing indicate zebrafish fear response and anxiety-like responses. Previous studies have shown the capability of zebrafish to show behavioral changes in the presence of sympatric predators such as Indian leaf fish (*Nandus nandus*) (Bass and Gerlai, 2008).

In this study, we examined zebrafish interspecies responses to eight different fishes purchasable in our area (Taiwan). The eight test fishes used in this study are: tiger barb (*Puntigrus tetrazona*), glass catfish (*Kryptopterus bicirrhis*), albino convict cichlid (*Amatitlania nigrofasciata*), black tetra (*Gymnocorymbus ternetzi*), bala shark (*Balantiocheilos melanopterus*), clown loach (*Chromobotia macracanthus*), Demason's cichlid (*Pseudotrophus demasoni*), and threadfin acara (*Acarichthys heckelii*). We also tested zebrafish responses to *A. nigrofasciata* in three different growth stages: adult, juvenile, and fry, and *G. ternetzi* in three different colors: red, yellow, and green.

Tiger barb was used in the role of negative control. It is an omnivorous tropical fish often kept as ornamental fish, native to central and southern Sumatra. Phylogenetically, it shares the same family with zebrafish (Cyprinidae) and was found to be a good tank mate for zebrafish (Stevens et al., 2017; Vasantharajan, 2023). In contrast, convict cichlid was used as a positive control. Native to Central America, this omnivore has previously been reported as a species capable of inducing fear in small fishes such as guppies and also has been used in zebrafish studies (Audira et al., 2018c; Goldman et al., 2018). Different convict cichlid growth stages were used to test zebrafish response to different sizes of visual stimuli, as previous studies have stated that the size of visual stimuli correlates with the fear response exhibited by zebrafish (Luca and Gerlai, 2012). Compared to these two fishes, the other fishes' behaviors were not well documented in the literature, however glass catfish, black tetra, clown loach, bala shark, and threadfin acara are known to be relatively peaceful, while Demason's cichlid are quite

[1]Department of Chemistry, Chung Yuan Christian University, Chung-Li 32023, Taiwan. [2]Department of Bioscience Technology, Chung Yuan Christian University, Chung-Li 32023, Taiwan. [3]Department of Biomedical Engineering, National Yang Ming Chiao Tung University, Taipei 112304, Taiwan. [4]Department of Biomedical Engineering, Chung Yuan Christian University, Chung-Li 32023, Taiwan. [5]Research Center for Aquatic Toxicology and Pharmacology, Chung Yuan Christian University, Chung-Li 32023, Taiwan.

*Authors for correspondence (sunbow@nycu.edu.tw; cdhsiao@cycu.edu.tw)

K.A., 0000-0002-3403-0879; G.A., 0000-0002-9985-6524; M.E., 0000-0003-0089-6516; C.-D.H., 0000-0002-6398-8672

aggressive and known to be territorial. All fishes tested in this study were omnivorous, with diets mostly consisting of aquatic invertebrates and plant matter. Most of them are native to Southeast Asia, except for Demason's cichlid, which is native to South Africa, and threadfin acara and black tetra, which are native to South America (Konings, 1994; Bogutskaya et al., 2008; Tan and Lim, 2008; Ng and Kottelat, 2013; Wahyudewantoro et al., 2022; Kadarini et al., 2025). Test fish behaviors are summarized in Table 1. The black tetra used in this study were genetically modified to exhibit different colors on their body, to observe the possibility of color preference of zebrafish through shared-environment tests (Siregar et al., 2020).

To study fear responses, previous studies have used zebrafish as an animal model, owing to their capabilities in showing fear or anxiety-like responses. Eight different fish species were used to test zebrafish behavioral response with *P. tetrazona* as negative control and *A. nigrofasciata* as positive control. To our knowledge, the other six fish species do not share any natural habitat with zebrafish and no interaction has been observed between these six species and zebrafish. Through the introduction of an individual fish to a group of zebrafish, we expected zebrafish to show behavioral phenotypes, fear or anxiety-like responses. For the eight fishes and their variations tested in this study, we hypothesized that there would be different degrees of behavioral response from the zebrafish depending on the test fish body size, as body size is consistently related to anti-predator behavior (Preisser and Orrock, 2012). To test our hypothesis, we conducted a shared-environment test, observing zebrafish behavioral responses (Audira et al., 2018a; Gerlai, 2020) and the movement trajectories of the test fish in 3D, using idtracker.ai, followed by behavioral endpoints calculation in Fish 3D Locomotion Analyzer (F3LA), and statistical analysis in GraphPad Prism (Fig. 1).

## RESULTS

### 3D zebrafish locomotor behavior altered after introduction of test fishes

A 3D locomotor-activity test was used to assess the alteration of zebrafish swimming behavior during the shared-environment tests. The 3D test was selected due to its sensitivity and comprehensiveness (Audira et al., 2018b) in observing locomotor activity endpoints. The introduction of several test fish to the environment altered the behavior of zebrafish: in the locomotor activity endpoints (Fig. 2A-D), the presence of green *G. ternetzi* reduced the average speed of zebrafish, while the presence of *P. demasoni* increased it (Fig. 2A). In the side view, more statistical differences were observed, zebrafish showed reduced average speed in shared-environment tests with *P. tetrazona*, *K. bicirrhis*, all *A. nigrofasciata*, and red and green *G. ternetzi*, while with *P. demasoni*, zebrafish showed increased average speed (Fig. S1A).

An increased swimming movement time ratio was observed in zebrafish in a shared-environment test with *K. bicirrhis*, juvenile *A. nigrofasciata*, and red and green *G. ternetzi*, while with *B. melanopterus*, *P. demasoni*, and *A. heckelii* the zebrafish showed a reduced swimming movement time ratio (Fig. 2B). Meanwhile, in the side view, an increase in zebrafish swimming movement was observed in a shared-environment test with red *G. ternetzi*. Reduced swimming movement was also observed in shared-environment tests with *B. melanopterus*, *P. demasoni*, and *A. heckelii*, similar to the 3D results (Fig. S1B).

An increased freezing movement time ratio was observed in zebrafish sharing environments with *B. melanopterus* and *A. heckelii*, while zebrafish sharing their environment with red *G. ternetzi* showed a reduction in freezing movements (Fig. 2C). From the side view observation, a significant increase in freezing

movement was observed in zebrafish exposed to *A. nigrofasciata*, green *G. ternetzi*, *B. melanopterus*, and *A. heckelii* (Fig. S1C).

The final locomotor endpoint observed was a rapid movement time ratio, which showed a similar pattern to average speed, with a reduced zebrafish rapid movement time ratio observed for the groups tested with *P. tetrazona*, *K. bicirrhis*, juvenile *A. nigrofasciata*, and green *G. ternetzi*, and an increased rapid movement time ratio for the group tested with *A. heckelii* (Fig. 2D). When observed from the side view, reduced rapid movement activity was present on zebrafish with *P. tetrazona*, *K. bicirrhis*, all *A. nigrofasciata*, and red and green *G. ternetzi*. Only zebrafish exposed to *P. demasoni* showed increased rapid movement observable from the side view (Fig. S1D).

Movement orientation endpoints included two minor endpoints: average angular velocity and meandering. Zebrafish angular velocity showed significant reduction in shared-environment tests with *P. tetrazona*, *K. bicirrhis*, and adult and juvenile *A. nigrofasciata*, and a significant increase during shared-environment testing with *B. melanopterus*, *P. demasoni*, and *A. heckelii* (Fig. 3A). Meandering movement was significantly increased in zebrafish exposed to fry *A. nigrofasicata*, all *G. ternetzi*, and *A. heckelii* (Fig. 3B). Movement orientation endpoint data were obtained only from the top view, angular velocity and meandering were not observed from side view, and the top view results were identical to the 3D results and are therefore not mentioned.

The exploratory behavior endpoints consisted of: average distance to tank center (thigmotaxis); total distance travelled at the top; number of entries to the top; time spent at the top; time spent in the middle; and time spent at the bottom. In the average distance to the center of the tank, we observed significantly higher tendency for the zebrafish to stay closer to tank center in shared-environment tests with *P. tetrazona*, K bicirrhis, juvenile and fry *A. nigrofasciata*, red *G. ternetzi*, *C. macracanthus*, and *B. melanopterus*, and an increased tendency to stay in the outer area when sharing their environment with *A. heckelii* (Fig. 4A). Meanwhile, if only side-view data were observed, zebrafish in a shared environment with *P. tetrazona*, *K. bicirrhis*, fry *A. nigrofasciata*, red *G. ternetzi*, or *C. macracanthus* showed a tendency to stay close to the tank center, while zebrafish tested with *A. nigrofasciata*, green *G. ternetzi*, and *A. heckelii* tended to stay farther away from tank center (Fig. S3A). The total distance zebrafish travelled in the top area was also significantly increased when they were in a shared environment with *A. nigrofasciata*, all *G. ternetzi*, *P. demasoni*, and *A. heckelii* (Fig. 4B). In the side view, zebrafish sharing their environment with all *A. nigrofasciata*, all *G. ternetzi*, *P demasoni* and *A. heckelii* travelled higher total distances in the top section of the tank, similar to the results obtained with 3D observations (Fig. S3B). Meanwhile, observed from top view, zebrafish showed increased tendencies to stay closer to the tank center when sharing environment with all of the test fish species (Fig. S4).

The number of entries zebrafish made to the top area was significantly reduced during shared-environment testing with *P. tetrazona*, *K. bicirrhis*, all *A. nigrofasciata*, all *G. ternetzi*, *C. macracanthus*, and *A. heckelii* (Fig. 4C). The time zebrafish spent in the top area was also increased during shared-environment testing with all *A. nigrofasciata*, all *G. ternetzi*, *P. demasoni*, and *A. heckelii* (Fig. 4D). Time spent in middle area by zebrafish was reduced during shared-environment testing with all *A. nigrofasciata*, yellow and green *G. ternetzi*, and *A. heckelii* (Fig. 4E). We also observed significantly reduced duration of stays in the bottom area by zebrafish during shared-environment testing with *K. bicirrhis*, all *A. nigrofasciata*, all *G. ternetzi*, *B. melanopterus*, *P. demasoni*, and *A. heckelii* (Fig. 4F). The results for number of entries to the top area,

**Table 1. Summary of zebrafish and test fish behavior**

| Species | Role in study | Family | Known behavior | Origin | Additional note | References |
|---|---|---|---|---|---|---|
| Zebrafish (*D. rerio*) | Observed species | Cyprinidae | - | South Asia | Common ornamental fish; omnivorous | Kalueff et al., 2013; Alderton, 2019 |
| Tiger Barb (*P. tetrazona*) | Negative control | Cyprinidae | Relatively peaceful; good tank mate for zebrafish | Southeast Asia | Common ornamental fish; omnivorous | Stevens et al., 2017; Alderton, 2019; Vasantharajan, 2023 |
| Convict cichlid (*A. nigrofasciata*) | Positive control | Cichlidae | Induces fear in small fishes (e.g. guppies, zebrafish); aggression varies with size | Central America | Previously used in zebrafish fear response study; omnivorous | Audira et al., 2018c; Goldman et al., 2018 |
| Glass catfish (*K. bicirrhis*) | Test species | Siluridae | Relatively peaceful; unknown interaction with zebrafish | Southeast Asia | Transparent body; omnivorous | Ng and Kottelat, 2013 |
| Black tetra (*G. ternetzi*) | Test species | Characidae | Relatively peaceful; unknown interaction with zebrafish | South America | Genetically modified with different body color; omnivorous | Bogutskaya et al., 2008 |
| Clown loach (*C. macracanthus*) | Test species | Bottidae | Relatively peaceful; unknown interaction with zebrafish | Southeast Asia | Omnivorous | Kadarini et al., 2025 |
| Bala shark (*B. melanopterus*) | Test species | Cyprinidae | Relatively peaceful; unknown interaction with zebrafish | Southeast Asia | Omnivorous | Wahyudewantoro et al., 2022 |
| Threadfin acara (*A. heckelii*) | Test species | Cichlidae | Relatively peaceful; unknown interaction with zebrafish | South America | Omnivorous | Tan and Lim, 2008 |
| Demason's cichlid (*P. demasoni*) | Test species | Cichlidae | Aggressive and territorial; possible aggressive behavior towards zebrafish | South Africa | Omnivorous | Konings, 1994 |

time spent at the top, time spent in the middle, and time at the bottom were identical between 3D observation and side-view only observation, indicating that no factors from the top view altered the values.

Fractal dimension was used to calculate the complexity of zebrafish movement during shared-environment testing. Zebrafish sharing their environment with *P. tetrazona*, juvenile *A. nigrofasciata*, green *G. ternetzi*, *B. melanopterus*, and *A. heckelii* showed reduced fractal dimension (Fig. 5A). In the side view, zebrafish with adult and juvenile *A. nigrofasciata*, green *G. ternetzi*, *B. melanopterus*, and *A. heckelii* showed reduced fractal dimension, while zebrafish with red *G. ternetzi* showed increased fractal dimension (Fig. S5A).

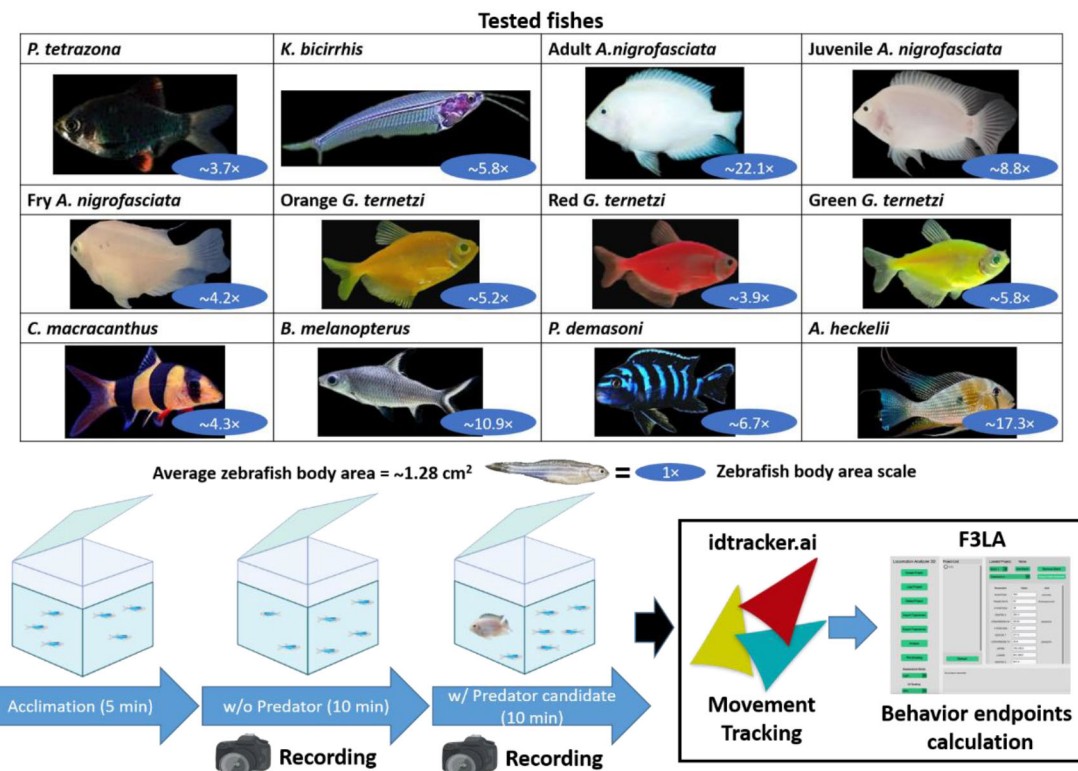

**Fig. 1. Fish species and their variance tested to induce a fear response in zebrafish (top) with average body area ratio relative to average zebrafish body area.** Study workflow encompassing zebrafish acclimation, recording without test fish, followed by recording of test fish species and their variants, movement tracking using idtracker.ai, and data analysis using Fish 3D Locomotion app (F3LA).

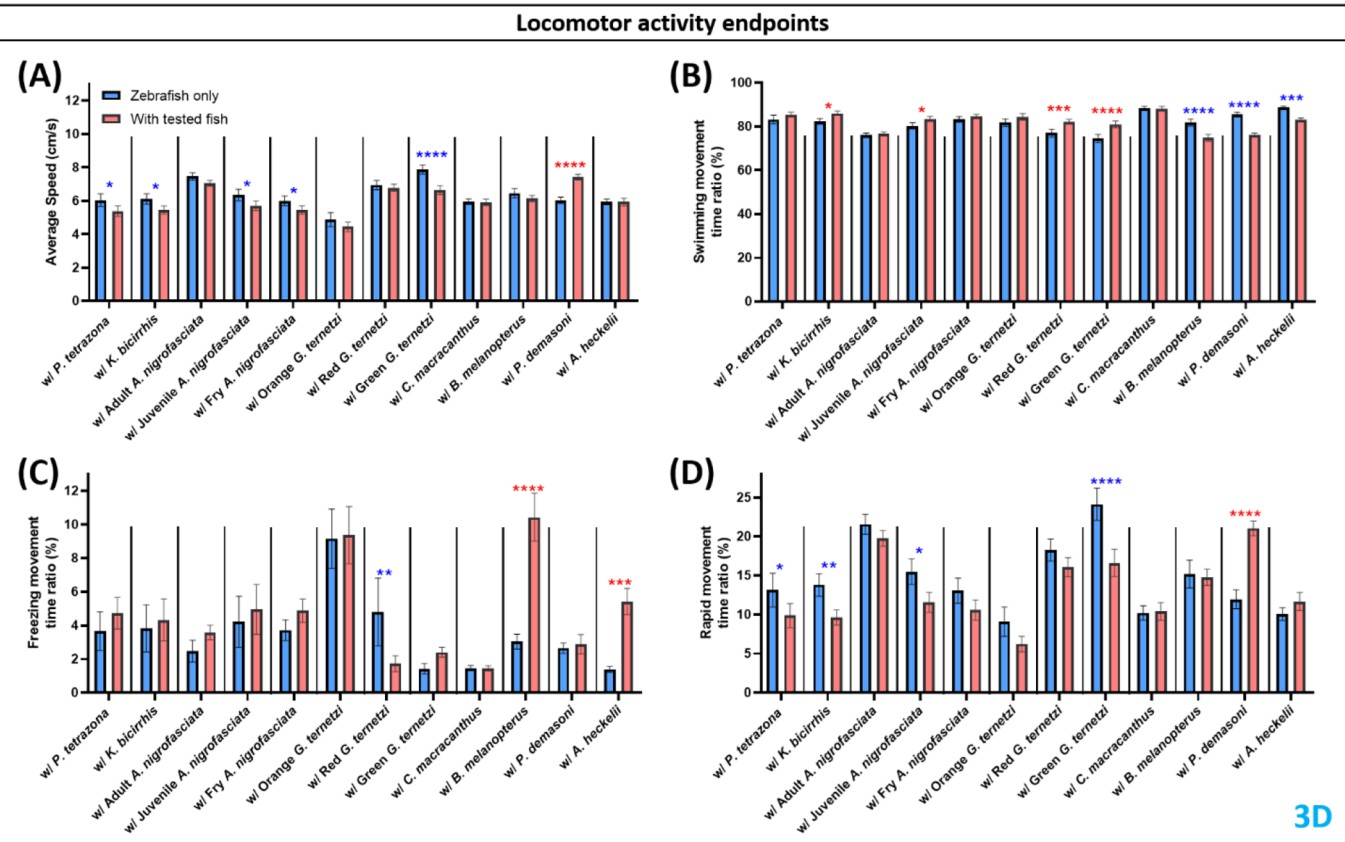

**Fig. 2. Comparison of zebrafish locomotor activity endpoints before (blue bar) and after introduction to tested fishes (red bar).** Four endpoints were calculated in this group, (A) average speed, (B) swimming movement time ratio, (C) freezing movement time ratio, and (D) rapid movement time ratio. Data are presented in a bar plot (mean±s.e.m.) and processed using two-way ANOVA mixed-effects analysis with uncorrected Fisher's LSD *post hoc* test (*n*=4, with six zebrafish per replication for zebrafish only and with test fish group; *P<0.05, **P<0.01, ***P<0.001, ****P<0.0001. Red asterisks represent increased activity when test fish was added to the tank in comparison to zebrafish only, blue asterisks represent decreased activity).

Observed from the top view, zebrafish fractal dimension appeared to increase during shared-environment tests with red and green *G. ternetzi*, and *C. macracanthus*, but reduced with *B. melanopterus* and *A. heckelii* (Fig. S6A).

Entropy was calculated to represent predictability of zebrafish movement during shared-environment testing. Zebrafish sharing their environment with *K. bicirrhis*, all *A. nigrofasciata*, orange and red *G. ternetzi*, and *C. macracanthus* showed reduced entropy,

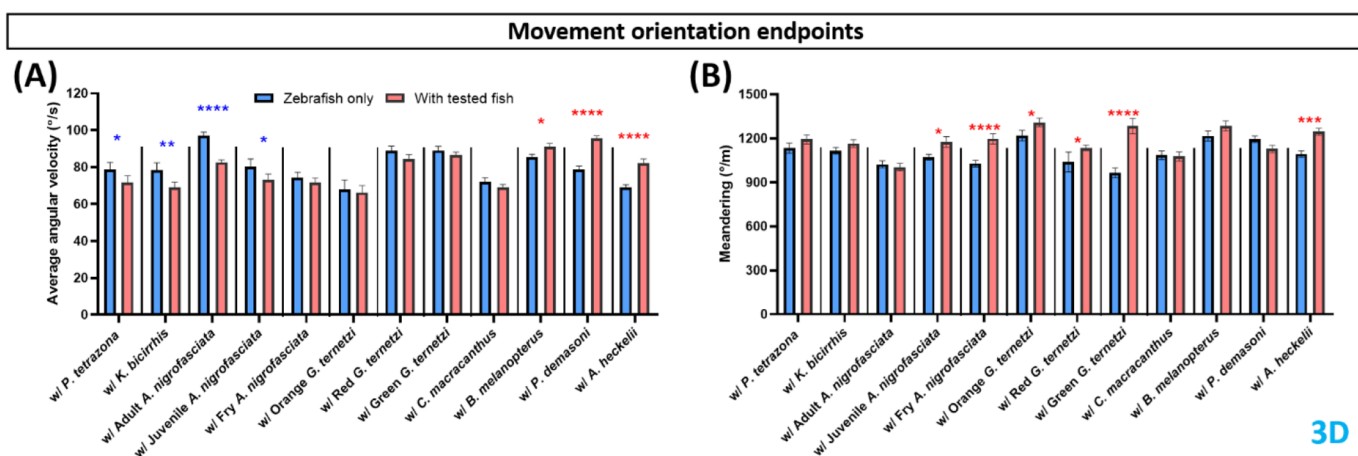

**Fig. 3. Comparison of zebrafish locomotor activity endpoints before (blue bar) and after introduction to tested fishes (red bar).** Four endpoints were calculated in this group, (A) angular velocity and (B) meandering. Data are presented in a bar plot (mean±s.e.m.) and processed using two-way ANOVA mixed-effects analysis with uncorrected Fisher's LSD *post hoc* test (*n*=4, with six zebrafish per replication for zebrafish only and with test fish group; *P<0.05, **P<0.01, ***P<0.001, ****P<0.0001. Red asterisks represent increased activity when test fish was added to the tank in comparison to zebrafish only, blue asterisks represent decreased activity).

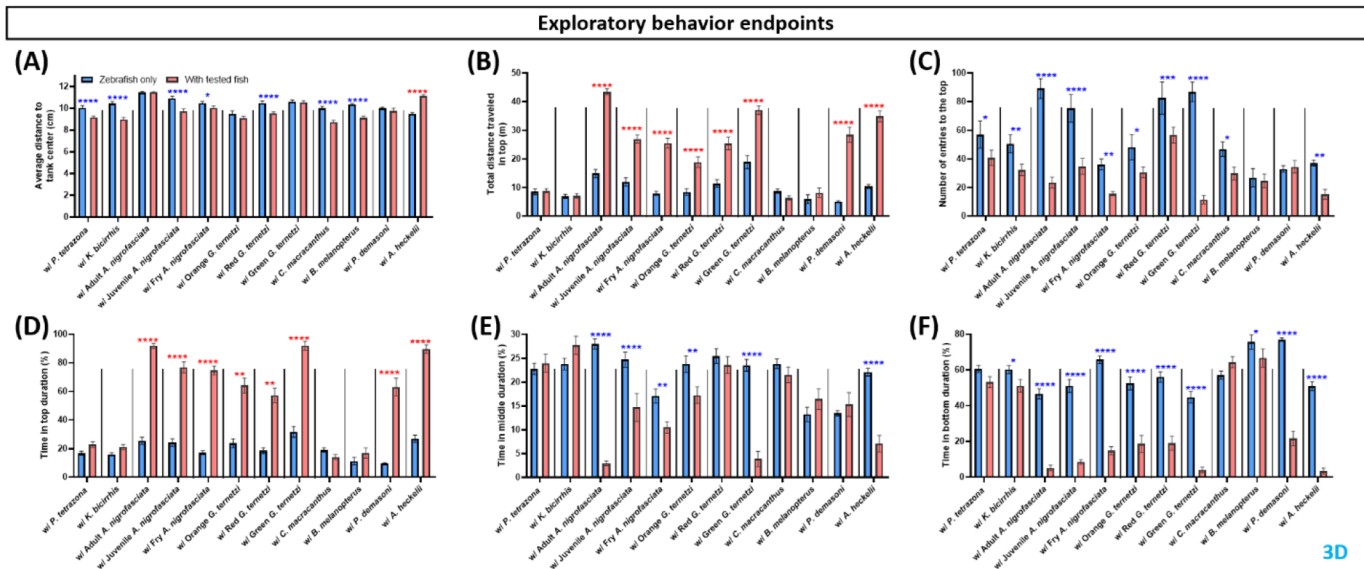

**Fig. 4. Comparison of zebrafish locomotor activity endpoints before (blue bar) and after introduction to tested fishes (red bar).** Four endpoints were calculated in this group, (A) average distance to tank center, (B) total distance traveled in top, (C) number of entries to the top, (D) time spent in the top area, (E) time spent in the middle area, and (F) time spent in bottom are. Data are presented in a bar plot (mean±s.e.m.) and processed using two-way ANOVA mixed-effects analysis with uncorrected Fisher's LSD post hoc test ($n$=4, with six zebrafish per replication for zebrafish only and with test fish group; *$P$<0.05, **$P$<0.01, ***$P$<0.001, ****$P$<0.0001. Red asterisks represent increased activity when test fish was added to the tank in comparison to zebrafish only, blue asterisks represent decreased activity).

while zebrafish with *B. melanopterus* and *A. heckelii* showed increased entropy (Fig. 5B). From the side-view observations, only zebrafish with adult *A. nigrofasciata*, and orange *G. ternetzi* showed increased entropy (Fig. S5B). In the top view, zebrafish entropy was reduced during shared-environment tests with *B. melanopterus* and *A. heckelii* (Fig. S6B).

Shoaling volume was also observed, since this is an important fear response in zebrafish. Shoaling volume was observed to be tighter during shared-environment tests with juvenile and fry *A. nigrofasciata*, green *G. ternetzi*, and *A. heckelii* (Fig. 5C). In the side view observations, shoaling area was reduced for zebrafish in a shared environment with juvenile *A. nigrofasciata* and green *G. ternetzi* (Fig. S5C), while top-view data showed tighter zebrafish shoaling areas in shared-environment tests with juvenile *A. nigrofasciata*, orange, and green *G. ternetzi* (Fig. S6C).

Afterwards, a Pearson's correlation test was conducted to investigate the correlation between test fishes' body size and zebrafish behavioral endpoints. From 16 tested endpoints, none of the endpoints showed very high correlation (0.7–1). Average distance to the center of the tank showed a high positive correlation (0.5–0.7) to body size at 0.6035. While moderate positive correlation (0.3–0.5) was observed on rapid movement time ratio, time in top duration, and average zebrafish to test-fish distance. Meanwhile, moderate negative correlation (−0.3–−0.5) was observed on time spent in the middle area, time spent at the bottom, and fractal dimension. Low positive correlation was observed on (0.1–0.3) on average speed, average angular velocity, total distance travelled in the top area, and entropy, while meandering, number of entries to the top, and shoal volume showed a low negative correlation (−0.1–−0.3). Finally, zebrafish freezing time ratio did not show any correlation (0–0.1) to the body size of the test fishes (Fig. S7, Table 2).

A further investigation into the correlation between test-fish body size and zebrafish behavior endpoints was conducted only on groups of zebrafish in shared-environment tests with *A. nigrofasciata* at different growth stages. The results showed that there are very high positive correlations between average speed, rapid movement time ratio, total distance travelled in the top section, and average distance between zebrafish and test fish. A very high negative correlation was observed in swimming movement time ratio. High positive correlation was observed between average angular velocity and average distance to the center of the tank and test-fish body area, while meandering and shoal volume showed high negative correlation to test-fish body area. Moderate positive correlation was observed on time in the top area and entropy, with time in the middle and time at the bottom showing moderate negative correlation to test-fish body area. Fractal dimension and freezing movement time ratio showed low negative correlation to body area, while fractal dimension showed no correlation to test-fish body area (Fig. S8, Table 2).

### Zebrafish-to-test fish avoidance, PCA, heatmap clustering in the presence of tested fishes, and parameters correlation in 3D and 2D viewpoints, from 16 tested endpoints

The distance of each zebrafish in the 3D tank to the test fishes was calculated. Statistically, zebrafish preferred to stay further from the *A. nigrofasciata* of different ages, green *G. ternetzi*, *P. demasoni*, and *A. heckelii* than they did from *P. tetrazona*, *K. bicirrhis*, orange and red *G. ternetzi*, and *B. melanopterus*. Meanwhile, zebrafish that shared an environment with *C. macracanthus* mostly kept their distance from them (Fig. 6). By observing zebrafish-to-test fishes distance measurement and the visualization of their swimming pattern (Fig. 7), we concluded that zebrafish avoid the former group of test fishes, but not the latter. Therefore, these results might indicate an avoidance response by zebrafish to the former group to a certain degree. Meanwhile, inside and top view data recorded a similar zebrafish-to-test-fishes distance profile in the groups with *P. tetrazona*, *K. bicirrhis*, red *G. ternetzi*, and *B. melanopterus*, shorter than that in the groups that included adult *A. nigrofasciata*, juvenile *A. nigrofasciata*, *P. demasoni*, and *A. heckelii*, which zebrafish avoid (Figs S9 and S10).

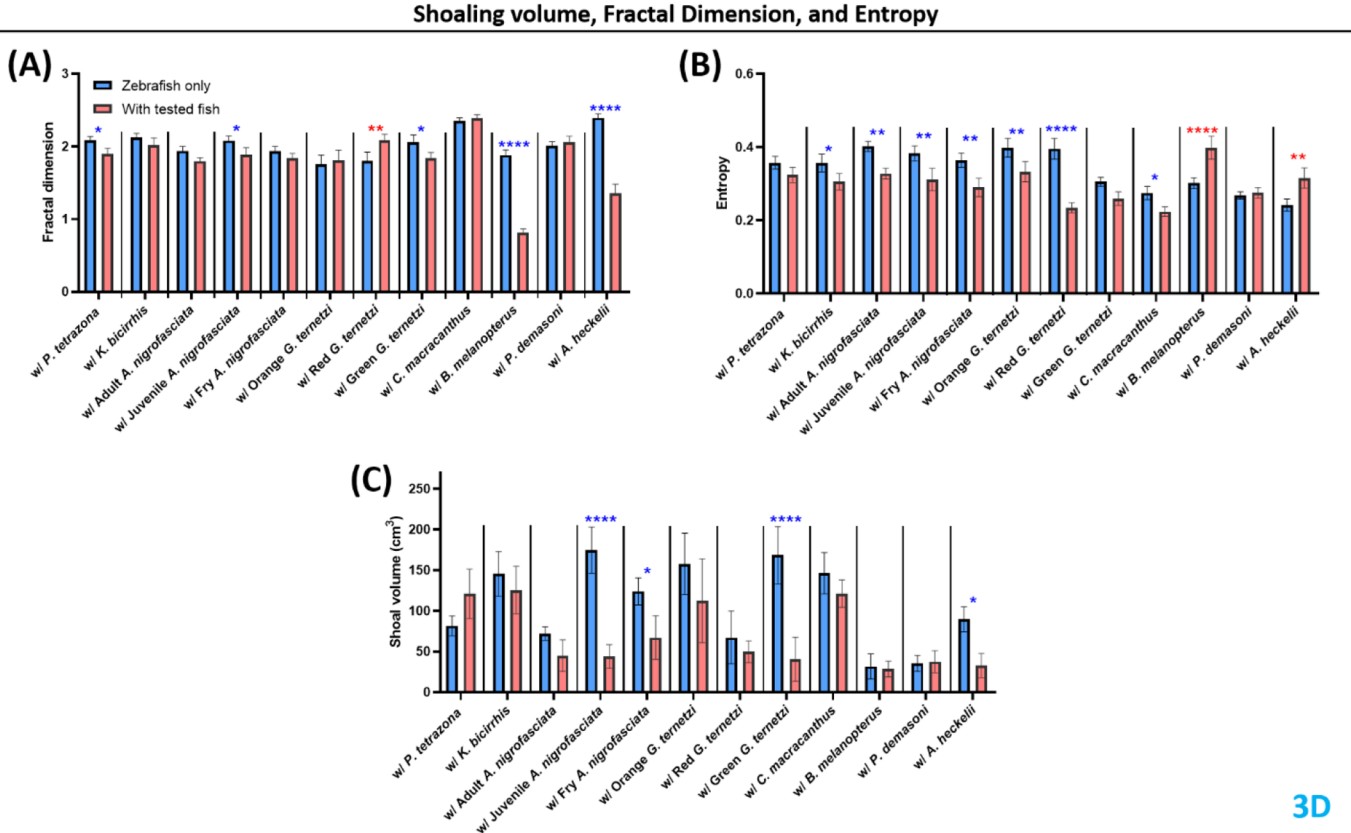

**Fig. 5. Comparison of zebrafish locomotor activity endpoints before (blue bar) and after introduction to test fishes (red bar).** Four endpoints were calculated in this group, (A) fractal dimension, (B) entropy, and (C) shoaling volume. Data are presented in a bar plot (mean±s.e.m.) and processed using two-way ANOVA mixed-effects analysis with uncorrected Fisher's LSD *post hoc* test (*n*=4, with six zebrafish per replication for fractal dimension and entropy endpoints, *n*=4 for shoaling; *P*<0.05, **P*<0.01, ****P*<0.001, *****P*<0.0001. Red asterisks represent increased activity when test fish was added to the tank in comparison to zebrafish only, blue asterisks represent decreased activity).

These results were further confirmed through PCA and heatmap clustering. The 3D result divided the test fishes into two major groups, the first group contained *P. tetrazona*, *K. bicirrhis*, orange and red *G. ternetzi*, and *C. macracanthus*; the second group included *A. nigrofasciata* of different ages, green *G. ternetzi*, *B. melanopterus*, *P. demasoni*, and *A. heckelii.* The PCA and heatmap results are quite similar to the zebrafish-to-test-fish distance, except in the case of *B. melanopterus* (Fig. 8). The heatmap and PCA for the side view are nearly identical to the 3D results, except in the case of *B. melanopterus*, which is grouped together with *P. tetrazona* (Fig. S11). PCA and heatmap of top view data showed more even more differences, *A. nigrofasciata*, which we used as positive control

**Table 2. Correlation (r value) between all test fishes and cichlid of various growth stages to zebrafish behavior endpoints**

| Endpoints | All tested fishes body area against zebrafish behavior endpoints correlation (r value) | Body area of cichlid in various growth stages against zebrafish behavior endpoints correlation (r value) |
|---|---|---|
| Average speed | 0.2704 | 0.7612 |
| Swimming movement time ratio | −0.4062 | −0.8725 |
| Freezing movement time ratio | 0.08963 | −0.1826 |
| Rapid movement time ratio | 0.3217 | 0.8296 |
| Average angular velocity | 0.2469 | 0.6070 |
| Meandering | −0.2326 | −0.6563 |
| Average distance to the center of the tank | 0.6035 | 0.6872 |
| Total distance travelled in the top | 0.2456 | 0.8784 |
| Number of entries to the top | −0.1691 | 0.0814 |
| Time in top duration | 0.3789 | 0.4914 |
| Time in middle duration | −0.3970 | −0.4296 |
| Time in bottom duration | −0.3026 | −0.4191 |
| Shoal volume | −0.2229 | −0.5202 |
| Average zebrafish to test fish distance | 0.3943 | 0.8291 |
| Fractal dimension | −0.3582 | −0.1694 |
| Entropy | 0.2791 | 0.3563 |

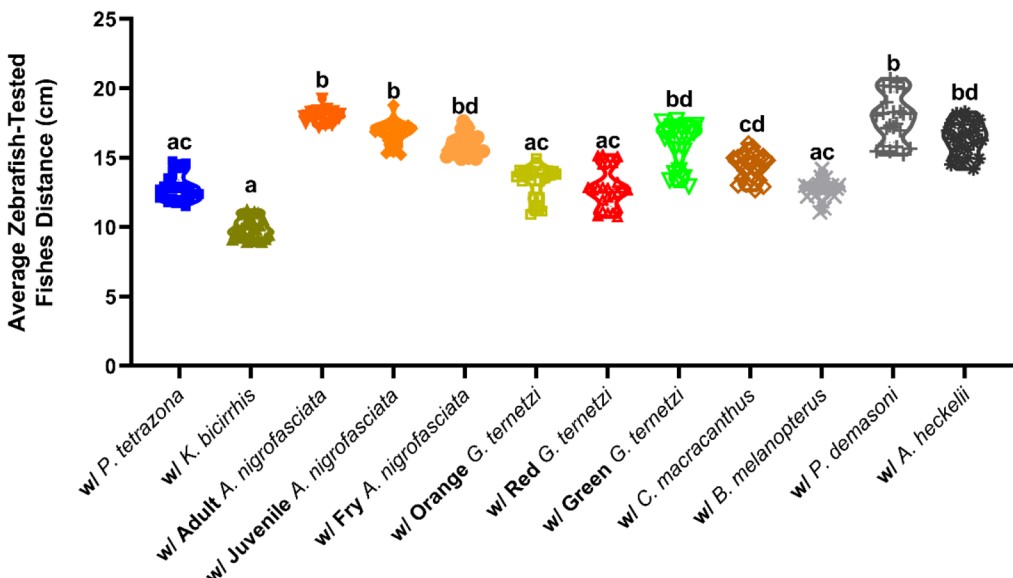

**Fig. 6. Comparison of average zebrafish to test fish distance in the water tank during 3D locomotion test.** Data are presented in a violin plot showing each point and processed using Kruskal–Wallis with Dunn's multiple comparison tests ($n=4$, with six zebrafish per replication; different letter represents statistically significant difference, $P<0.05$).

to induce zebrafish avoidance, was grouped together with *P. tetrazona*, the negative control. In addition, green *G. ternetzi* is also in the same group, while orange *G. ternetzi* is in the other group (Fig. S12). An additional test using Pearson correlation was conducted to observe the correlation between 3D, 2D-top-view, and 2D-side-view data. From nine tested endpoints, only rapid movement time ratio and zebrafish to test fish distance showed a strong correlation between 3D, 2D-top-view, and 2D-side-view data (Fig. 9).

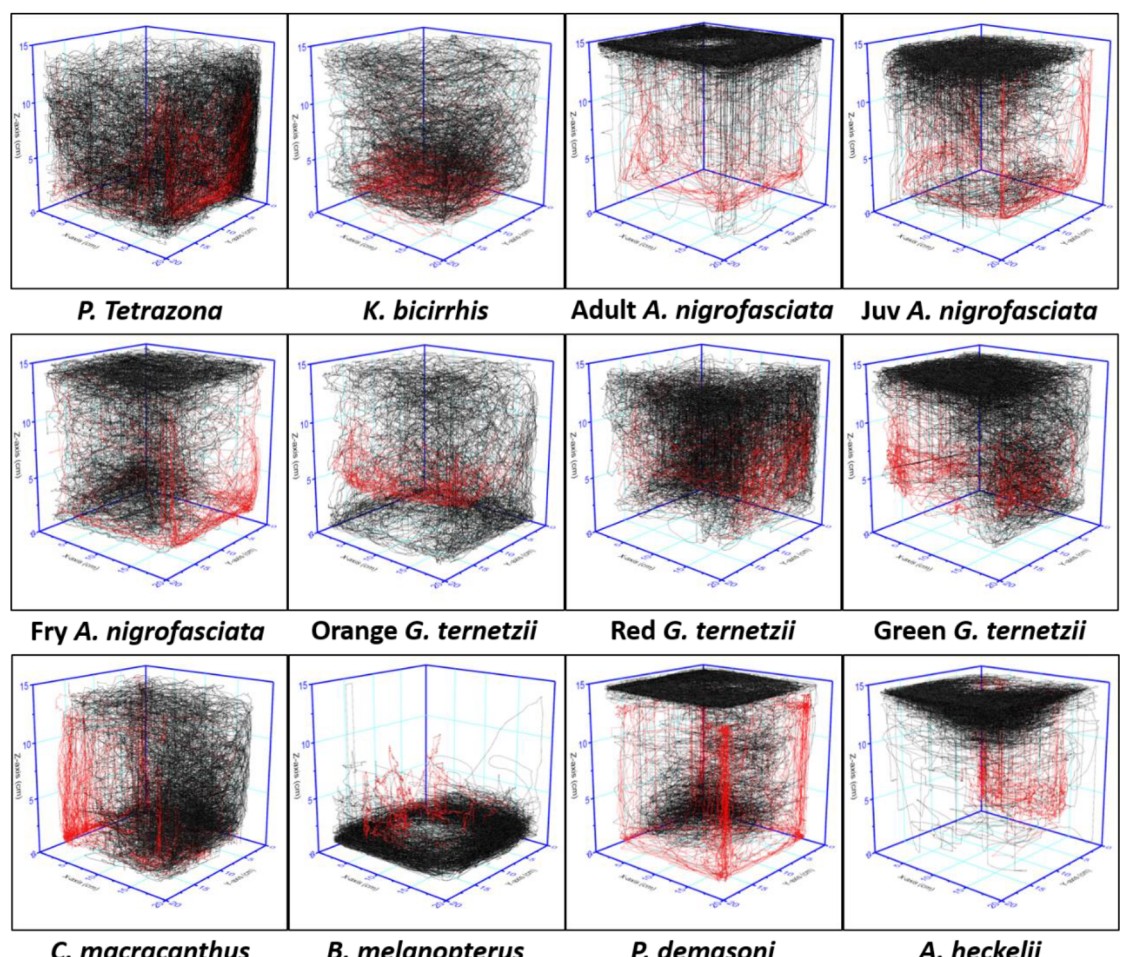

**Fig. 7. Visualization of six representative zebrafish (black) and tested fish (red) swimming patterns during a shared-environment test (10 min) in a 3D tank.** Tank was filled to 75% of maximum volume with water [final dimension of 20 cm (X-axis) ×20 cm (Y-axis) ×15 cm (Z-axis)].

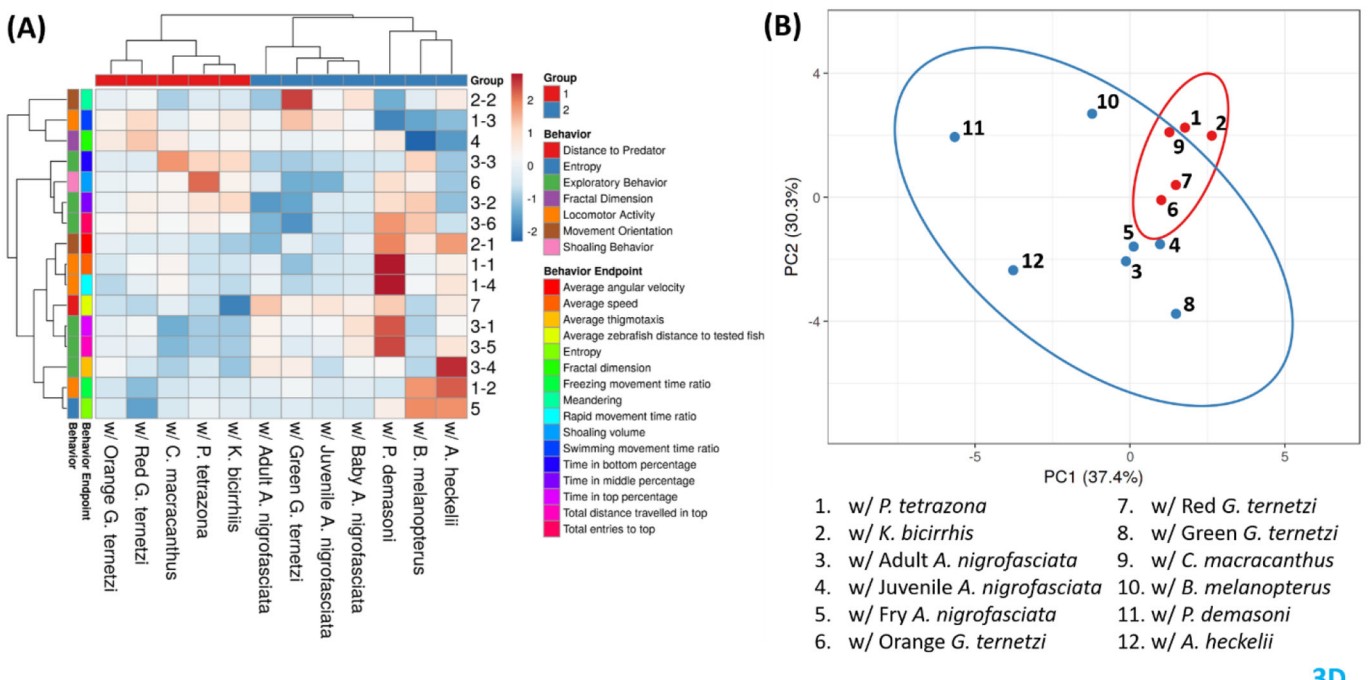

**Fig. 8. (A) Heatmap clustering analysis and (B) principal component analysis (PCA) of all zebrafish behavioral endpoints in 3D after introduction to test fish.** Two major clusters were formed from the heatmap clustering marked with red (group 1) and blue (group 2).

## DISCUSSION

The results from our findings allowed us to distinguish test fishes capable of inducing fear in zebrafish from those that are not. According to PCA and heatmap results, *P. tetrazona* and *A. nigrofasciata* are grouped separately, this result supports our hypothesis regarding the response of zebrafish to these two fishes. In the presence of *P. tetrazona*, zebrafish stay closer to the tank center and move slower than controls (lower average speed and rapid movement time ratio). Vertical exploration was found to be unaffected in the three sections of the tank (top, middle, and bottom) compared to zebrafish-only environments, which means that *P. tetrazona* do not induce anxiety-like or predator-avoidance responses in zebrafish (Colwill and Creton, 2011). A shared-environment test with *K. bicirrhis* also showed similar behavior alteration to *P. tetrazona*. While *K. bicirrhis* body size is generally bigger than *P. tetrazona* (Alderton, 2019), its presence seems to be less intimidating to zebrafish, which was unexpected, considering the common beliefs about their social dominance (Alcazar et al., 2014). This difference might be due to the body transparency of *K. bicirrhis* creating the illusion of smaller body size. Another fish being grouped with *P. tetrazona* is *C. macracanthus*. During shared environment test with *C. macracanthus*, zebrafish was found to be mostly unaffected by their presence, except for a reduction in entropy.

Previous studies have reported zebrafish to show a preference toward certain colors, in the environment this preference is related to defensive mechanisms, species recognition, and foraging behavior. This color preference might be related to fear response due to its connection to a defensive mechanism, while also being trained from experience and existing innately. Zebrafish eyes are known to be capable of perceiving ultraviolet, blue, green, and red (Spence and Smith, 2008). To test this, we used three *G. ternetzi* with different body colors. While exposure to all three *G. ternetzi* colors led to an increase of zebrafish time in the top area followed by a reduction of time spent in the middle and bottom, exposure to green *G. ternetzi*

led to increases in time spent in the top area compared to the other colors. The average distances between zebrafish and test fish are also different between green, red and orange *G. ternetzi*, with zebrafish tending to stay farther from green *G. ternetzi* than to red and orange *G. ternetzi*, thus the significant difference shown in this endpoint. Zebrafish also shoal more tightly during exposure to green *G. ternetzi*, while shoal volumes were unchanged during exposure to red and orange *G. ternetzi*. These results are further supported by the PCA and heatmap results, where red and orange *G. ternetzi* is grouped with *P. tetrazona*, while green *G. ternetzi* is grouped with *A. nigrofasciata*. This preference had been mentioned in a previous study where zebrafish behavior is more active when exposed to light at a medium wavelength (480-540 nm), compared to a high wavelength or short wavelength (Risner et al., 2006). In this case, green *G. ternetzi* correlates to the medium wavelength at 491-575 nm, while orange and red to wavelengths at 585-700 nm (Orna, 2012; Park et al., 2016).

While *A. nigrofasciata* has been proven to be an alternative fear inducer in zebrafish studies (Audira et al., 2018b), to our knowledge, no study has examined zebrafish reactions to *A. nigrofasciata* at different growth stages. Therefore, we tested *A. nigrofasciata* at three different growth stages: adult, juvenile, and fry. Sharing the same environment with adult and juvenile *A. nigrofasciata* reduced zebrafish average angular velocity significantly. Zebrafish meandering was not altered during shared-environment testing with adult *A. nigrofasciata* but significantly reduced in shared-environment tests with juvenile and fry *A. nigrofasciata*. Furthermore, the presence of all *A. nigrofasciata* made zebrafish stay mostly in the top area of the tank (Fig. 4), while average zebrafish to test fish distance was similar but slightly different depending on their growth stages (Fig. 6) and their overall movement pattern (Fig. 7), in which the presence of adult *A. nigrofasciata* made them stay in the top area, with specific diving behavior related to shoaling zebrafish as a defense mechanism in the presence of a predator

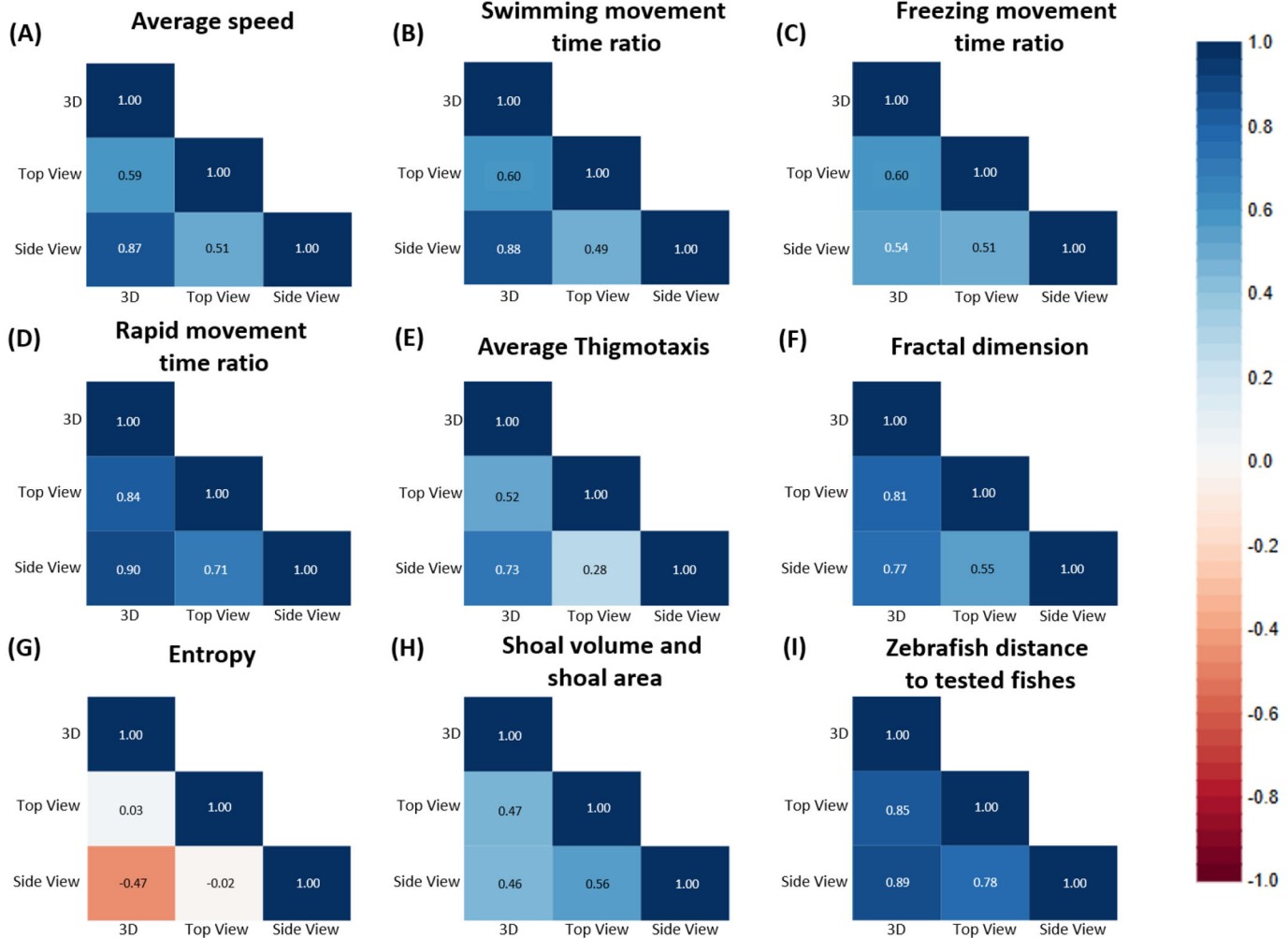

**Fig. 9. Pearson correlation of (A) average speed, (B) swimming time ratio, (C) freezing time ratio, (D) rapid movement time ratio, (E) average thigmotaxis, (F) fractal dimension, (G) entropy, (H) shoal volume and shoal area, and (I) zebrafish distance to test fish of 3D and 2D (top and side view) data.**

(Kalueff et al., 2013). It was commonly believed that the zebrafish would stay in the bottom of the tank due to fear, but our results indicate that zebrafish respond to *A. nigrofasciata* by staying in the top area of the tank, which might be attributed to *A. nigrofasciata* preferring to stay at the bottom of the tank. This result is similar to previous study (Ladu et al., 2015), which used *Astronotus ocellatus*, an allopatric zebrafish predator that tends to remain at the bottom of the tank, increased zebrafish staying time in top area significantly, despite the presence of separator between zebrafish and *A. occellatus*.

*B. melanopterus* is the next fish grouped with *A. nigrofasciata* in the PCA result. During the shared environment test, zebrafish showed increased freezing behavior, with no reduction in average speed and an increase in angular velocity. In the exploratory behavior endpoints, there is an increase of zebrafish time spent at the bottom and tendency to stay closer to the tank center. Additionally, it was observed that the zebrafish did not try to avoid *B. melanopterus*, since the distance from the test fish is more similar to *P. tetrazona* than to *A. nigrofasciata*. This result is probably due to the increased freezing behavior, instead of avoidance attempts made by zebrafish, which might indicate an anxiety-like behavior from zebrafish towards this species.

We were able to observe the fear response from zebrafish sharing an environment with *P. demasoni*. During the shared-environment

test, zebrafish showed higher avoidance of *P. demasoni* than *A. nigrofasciata*, displayed by the increases in average speed, rapid movement, and angular velocity. Similar top area preference and tightened shoaling were also observed when sharing an environment with *P. demasoni*, with similar average distances from zebrafish to test fishes as those seen in shared-environemnt tests with *A. nigrofasciata*. However, from the 3D projection, *P. demasoni* seems to be more active than *A. nigrofasciata* (Fig. 7).

Finally, during the shared-environment test with *A. heckelii*, zebrafish displayed increased freezing behavior, average angular velocity and meandering. In the exploratory behavior endpoints, zebrafish showed an increased preference to stay on the top area of the tank and keep their distance from *A. heckelii*, similar to behaviour observed in shared-environment tests with *A. nigrofasciata*.

The 3D observation result showed numerous differences from the 2D observation. The most important advantage to 3D observation is that more important endpoints can be encompassed, compared to observing the behavior only from one viewpoint at a time. For example, meandering and angular velocity can only be observed from the top view and vertical area preference can only be observed from the side view. Additionally, it is also possible to reconstruct 3D trajectories to improve overall endpoint accuracy compared to only using one viewpoint (Lin et al., 2024).

Body area correlation test of all test fish showed from 17 tested endpoints only average distance to the center of the tank have high correlation between zebrafish response to tested fishes body size, eight endpoints showed moderate correlation, seven endpoints showed low correlation, and one endpoint showed no correlation. Another body correlation analysis was conducted in smaller scope, by only observing *A. nigrofasciata* of different growth stages. In this study, three *A. nigrofasciata* growth stages were used: adult, juvenile, and fry, with body-area ratios of 22.1×, 8.8×, and 4.2× respectively, in comparison to zebrafish. From this test, five endpoints showed very high correlation, four endpoints showed high correlation, five endpoints showed moderate correlation, two endpoints showed low correlation, and one endpoint showed no correlation. From these results, we conclude that zebrafish fear response depends on the species of the tested fishes, and higher antipredatory responses can be observed towards fish of the same species with bigger body sizes. This zebrafish fear response was similar to that seen in a previous study on a different fish species, *Poecilia mexicana*. In the study, *P. mexicana* was exposed to four different fish species and showed the highest predator avoidance to fish that are more active, and possibly to common morphological features other than the body size (Bierbach et al., 2013; Alcazar et al., 2014).

From the behavioral test results, we infer that *P. demasoni* and *A. heckelii* are comparable to *A. nigrofasciata* in their ability to induce fear responses in zebrafish. Both of these fishes showed increased avoidance by zebrafish, caused zebrafish to stay in the top area of the tank in order to avoid them, and made zebrafish stay closer to each other. The difference is only apparent in locomotor activity endpoints, in which *P. demasoni* increased zebrafish rapid movement, while *A. heckelii* increased the freezing movement of zebrafish. Green *G. ternetzi* also altered zebrafish behavior in a similar way to *A. nigrofasciata* to a lesser degree. On the other hand, zebrafish response to *B. melanopterus* seems unique, high freezing and meandering might be a sign of anxiety. Previous studies have stated the difficulty in identifying the difference between fear and anxiety. It was hypothesized that only fear was observed in this study as the fishes was introduced abruptly to the zebrafish environment, due to the nature of fear as a response to immediate threat, but there is a possibility that anxiety can be shown in this study (Gerlai, 2020). However, we cannot ascertain anxiety from fear in zebrafish response to *B. melanopterus* as further biochemical analysis is necessary. Another important thing to highlight is that, with the exception of green *G. ternetzi*, fish species capable of inducing fear response through avoidance behavior in zebrafish were from the Cichlidae family. A previous study had also reported the use of pike cichlid (*Crenicichla alta*) (Dalton and Flecker, 2014) as a predator in a study on fear response in guppies. These results suggest that fish from Cichlidae family might be an effective fear inducer for guppies, zebrafish, or other small fishes, but this will require a more detailed study in the future, using Cichlid of different species or using inanimate object (moving robot or videos) designed with morphological features of Cichlids in mind. The design of test tank might also not be ideal in this study as several zebrafish behavioral endpoints might be harder to interpret due to direct stimulation from the test fish, which can be overcome through the use of a transparent barrier with perforations to apply both visual and chemical stimuli from the tested species while reducing the complexity of zebrafish response. Additionally, the zebrafish gender ratio in this study were random and might also come as a limitation of the presented results. As there have been studies that showed different zebrafish behavior depending on the gender ratio within the group (Ruhl et al., 2009), which can also be improved in the future study along with previous limitations.

Zebrafish has proven itself to be a good model for studying behavioral phenotypes of fear and anxiety, with their ease of maintenance, low costs, and high fecundity. This makes them a preferred animal model for fear and anxiety-related studies. This study has highlighted the possibility of an alternative fear inducer such as *P. demasoni* and *B. melanopterus* in inducing fear to induce zebrafish fear responses, the relation of body size to the intensity of zebrafish behavioral response, zebrafish color preference through different *G. ternetzi* GloFish variance, and the importance of 3D observation to avoid dimensional reduction.

This study on zebrafish behavioral responses on eight different tropical fishes through 3D observation shows behavioral differences depending on the test species and their variance. We highlight several important endpoints to measure fear through 3D observation, such as distance to tested fishes, vertical area preference, followed by an increase in rapid, freezing, and erratic movement (meandering), similar to the previous study (Gerlai, 2020). While the previous study stated that zebrafish might stay in the bottom area in response to fear, our study showed top area preference in response to the presence of some test fishes, which might be an escape response as some of the tested fishes tend to stay in the bottom area of the tank. We also conclude that the use of 2D observation might be insufficient, as several endpoints are only observable from a specific viewpoint, therefore 3D observation is recommended to obtain the desired endpoints and avoid dimensional reduction (Lin et al., 2024). The test on the correlation of fishes body size to zebrafish behavioral response suggested that there is no correlation between body size of tested fishes to fear response by zebrafish, but by reducing the scope of observation, a correlation between test-fish body-area size and zebrafish fear response was observed in *A. nigrofasciata*, however, it must be taken into account that we did not observe the behavioral differences in zebrafish response to test fish at different growth stages. We are also able to display zebrafish color preference by using *G. ternetzi* of different body colors, with zebrafish showing preference for yellow and red *G. ternetzi* and avoiding green *G. ternetzi*. Finally, we are able to find two important fishes; *P. demasoni* and *A. heckelii*, that are able to induce similar fear response from zebrafish to *A. nigrofasciata*, and we show that *B. melanopterus* induce an anxiety-like response from zebrafish through behavioral phenomics.

## MATERIALS AND METHODS
### Animal maintenance
Mixed-gender adult zebrafish aged at least 3 months old (∼3 cm body length) were purchased from a local aquarium vendor, Zgenebio Inc. in Taipei, Taiwan (https://www.zgenebio.com.tw/english.html, accessed on 5 September 2024). Purchased zebrafish were kept in a 110 cm×35 cm×40 cm holding tank with a stock density of 3.6 fish per liter. Zebrafish were acclimatized prior to the study for at least 1 month after arrival. Zebrafish were fed using Artemia and frozen food (alternate daily) twice per day, in the morning and afternoon, maintained in a continuously aerated aquarium system at 26±1°C with 14:10 h light:dark cycle, and water pH at 7.0-7.5. The different fishes used in this study were also kept in identical condition as the zebrafish, but in a smaller holding tank with a size of 47.5×33.5×25 cm with a stock density ranging from 0.3 to 2 animals per liter depending on each fish's body size. The fishes were Tiger barb (*Puntigrus tetrazona*), Glass catfish (*Krypytopterus bicirrhis*), adult, juvenile, and fry convict cichlid (*Amatitlania nigrofasciata*), orange, red, and green Black tetra (*Gymnocorymbus ternetzi*), Clown loach (*Chromobotia macracanthus*), Bala shark (*Balantiocheilos melanopterus*), Demason's cichlid (*Pseudotropheus demasoni*), and Threadfin acara (*Acarichtyhs heckelii*) purchased from local aquarium vendor.

## 3D zebrafish interspecies interaction recording setup

The recording setup to observe the interaction between zebrafish and other fishes was prepared according to a previous publication using an acrylic 20×20×20 cm fish tank filled with fish reservoir water to 75%. A mirror was attached above the fish tank at a 45° angle to reflect the top view of the tank. A custom light-emitting diode (LED) platform (by ZGene Biotech Inc., http://www.zgenebio.com/) was used as a background light source, emitting light from behind and below the fish tank, additionally, non-transparent covers were attached to the left and right sides of the tank to minimize distraction. Videos were recorded at 1280×720 pixels using a Canon EOS 600D DSLR camera with a zoom lens (EF-S 55-250 mm, Canon), placed ~6 m in front of the fish tank. The experiment was conducted in the afternoon 12:00 h-16:00 h (Audira et al., 2018a). Six naïve adult zebrafish in a mixed gender with a randomly sampled male/female ratio were gently moved from the holding tank to the observation tank using a fishnet for observation. Zebrafish were acclimated in the water tank for at least 5 min before recording. After the acclimation period, the zebrafish were recorded for 10 min as the control group, then a naïve fish from the other species was introduced to the tank and the recording was immediately started for 10 min. The recording was done in quadruplicate for tested fishes, using six naïve zebrafish and one test fish for each replication. Totaling to 24 zebrafish and four fish per test-fish species, with the addition of four fish for each growth stage of *A. nigrofasciata* and four fish with different body colors for *G. ternetzi*. Afterward, zebrafish and test fish movement was tracked using idtracker.ai according to the idtracker.ai user guide (Romero-Ferrero et al., 2019). The movement tracking was conducted separately for the top view and side view of the tank and merged using F3LA.

## 3D locomotion data processing using Fish 3D Locomotion Analyzer (F3LA)

F3LA is a self-built Python-based app, used to assist in coordinate matching, data processing, and endpoints calculation processes from the idtracker.ai results. The application matches the coordinates of each individual from different viewpoints (top and side view) to a 3D coordinate, which is used to calculate the locomotor activity endpoints (comprised of average speed, swimming time movement ratio, freezing time ratio, and rapid movement time ratio), movement orientation endpoints (comprised of angular velocity and meandering), exploratory behavior endpoints [comprised of average distance to tank center (thigmotaxis), time in top duration, total distance traveled in top area, time in middle area, number of entries to the top, and time in bottom duration], shoaling volume, fractal dimension, and entropy (Luong et al., 2024). Additionally, we also observe the average inter-zebrafish distance and distance of each zebrafish in the tank to the other fishes by calculating their centroid distance. Information regarding measured endpoints and their descriptions is summarized in Table S1 (Audira et al., 2018a).

## Statistical analysis

The statistical analysis and result graphs plotting were conducted in GraphPad Prism (GraphPad Software version 8 Inc., La Jolla, CA, USA). Two-way ANOVA mixed-effects analysis was used to test the effect of test fish addition to zebrafish behavioral phenotypes during shared-environment testing, followed by uncorrected Fisher's LSD post hoc test. Kruskal–Wallis with Dunn's multiple comparison test was used to compare the distance between zebrafish and test fish during shared-environment tests. The statistical difference between control and treated groups were indicated with *$P<0.05$, **$P<0.01$, ***$P<0.001$, ****$P<0.0001$. The body size of each fish was measured using ImageJ FIJI Build v1.52j (https://imagej.net/software/fiji/downloads, accessed on 20 September 2024), and the correlation between the body area of each different fish and F3LA-processed zebrafish behavior endpoints (all locomotor activity endpoints, all movement orientation endpoints, all exploratory behavior endpoints, shoaling volume, fractal dimension, and entropy) was calculated using Pearson correlation test. Additionally, a correlation matrix was also calculated to observe the relationship between 3D and 2D viewpoints using DisplayR (https://www.displayr.com/, accessed on 12 January 2025)

## Principal component analysis (PCA) and heatmap clustering analysis

ClustVis (https://biit.cs.ut.ee/clustvis/) (Metsalu and Vilo, 2015) was used to do the PCA and heatmap clustering analysis. PCA and heatmap analysis were used to obtain a more comprehensive visualization of behavior results by reducing the dimensionality of the dataset and thus, enabling us to explore the behavioral phenomics changes between zebrafish group during exposure to other fishes and variation tested in this study. Before the analysis in ClustVis, a comma-separated value (.csv) file containing the mean value of every calculated zebrafish behavior endpoint obtained from F3LA was summarized using Microsoft Excel. There are no transformations applied to the data and singular value decomposition (SVD) was selected as the imputation method. The PCA and heatmap results were then downloaded and saved in the personal system.

## Ethical statement

The care and use of experimental animals complied with the Institutional Animal Care and Use Committee (IACUC) at Chung Yuan Christian University animal welfare laws, guidelines, and policies as approved by Approval No.112010, issued on 29 December 2022.

### Acknowledgements

We would like to thank the Aquatic Toxicology and Pharmacology Service Platform, Center for Nanotechnology of Chung Yuan Christian University for providing the workplace and animal housing for this study.

### Competing interests

The authors declare no competing or financial interests.

### Author contributions

Conceptualization: K.A.K., C.-D.H.; Data curation: G.A.; Formal analysis: K.A.K., G.A., M.E.S.; Funding acquisition: C.-D.H.; Investigation: T.-R.G.; Methodology: M.E.S.; Supervision: T.-R.G., C.-D.H.; Writing – original draft: K.A.K., C.-D.H.

### Funding

Funding was supported by the National Science and Technology Council (113-2313-B-033-001-MY3). Open Access funding provided by National Science and Technology Council, Taiwan. Deposited in PMC for immediate release.

### Data and resource availability

All relevant data and details of resources can be found within the article and its supplementary information.

### Peer review history

The peer review history is available online at https://journals.biologists.com/bio/lookup/doi/10.1242/bio.062110.reviewer-comments.pdf

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
