## [Peer Review File · Biology Open]

Comparative analysis of Zebrafish fear responses to eight different fish species using 3D locomotion tracking assays

Kevin Adi Kurnia, Gilbert Audira, Michael Edbert Suryanto, Tzong-Rong Ger and Chung-Der Hsiao

DOI: 10.1242/bio.062110

Editor: Sandhya Koushika

Review timeline

Original submission:	10 June 2025
Editorial decision:	19 June 2025
First revision received:	8 September 2025
Accepted:	11 September 2025

Original submission

First decision letter

MS ID#: bio.062110

MS Title: Comparative analysis of Zebrafish fear responses to eight different fish species using 3D locomotion tracking assays

Authors: Kevin Adi Kurnia, Gilbert Audira, Michael Edbert Suryanto, Tzong-Rong Ger and Chung-Der Hsiao

I have now reached a decision on the above manuscript.

The reviewer reports are shown at the bottom of this email or can be accessed, together with a copy of this decision letter, by going to:

As you will see, the reviewers raised a number of substantial criticisms that prevent me from accepting the paper at this stage.

A substantially revised version might prove acceptable, if you can address their concerns. Of particular concern how one can separate zebrafish response to test fish stimulus per se versus how zebrafish may be responding to the behavior and interactions with the test fish since they are in the same tank. Additional experiments may be necessary to address this criticism. If you think that you can deal satisfactorily with the criticisms on revision, I would be pleased to see a revised manuscript. We would then return it to the reviewers.

At this stage, we also ask you to ensure your manuscript complies with our formatting guidelines. Provided you are able to fully address the referees' comments, we are positive about publication of your paper (we accept over 95% of revision submissions) and therefore hope you won't mind any extra work involved in reformatting your manuscript at this point.

Please ensure that you clearly highlight all changes made in the revised manuscript. Please avoid using 'Tracked changes' in Word files as these are lost in PDF conversion.

I should be grateful if you would also provide a point-by-point response detailing how you have dealt with the points raised by the reviewers in the 'Response to Reviewers' box. Please attend to

all of the reviewers' comments. If you do not agree with any of their criticisms or suggestions please explain clearly why this is so.

Reviewer 1

Comments for the author

Kurnia and colleagues describe a series of experiments where they attempt to determine how groups of zebrafish respond to different fish species. While an interesting idea, there are some significant flaws in the experimental design that dampen my enthusiasm. There is also some sloppiness in the figures and captions and the overall writing also needs to be improved.

Minor points

Line 39, Fear is not a 'sensor', but a response to perceived threat.

Line 51, this is awkwardly written, zebrafish don't elicit responses to anxiety. They elicit anxiety like responses.

Line 54, this is awkwardly written, freezing is not related to zebrafish response to fear. It is a response indicative of fear.

Line 57, citation?

Line 58, 'we would like to observe' is incorrect. "We examined zebrafish interspecies responses..."

Lines 59-66. Are these fish natural predators of zebrafish? Does their natural range overlap with those of zebrafish? Why would it be expected that zebrafish would respond fearfully to some fish species versus others?

Line 179, the word 'palpable' here is inappropriate.

Line 566. Should be 'Fish species and their variants used...'

I would suggest including a table of the different fish species used, their native habitats, and relevant characteristics.

Line 102, units on the size of the holding tank

Line 109-114: What is the source of other 'test' fish?

Line 318-319: It is inappropriate to state that zebrafish are relaxed and confident in the presence of tetrazona. Please only describe the behavior without undue anthropomorphism.

Line 321-322: Staying near the top of the tank is not usually interpreted as 'increased confidence'. Increased confidence in what? It has been interpreted as a reduction in anxiety-like behavior or more willingness to take risk. However, I don't know that you can make this interpretation in your data. The issue is that the behavior of the 'test' fish is unaccounted for. If the test fish is swimming near the bottom of the tank and preventing the zebrafish from spending time there, does that make the top dwelling behavior a sign of decreased anxiety or increased risk taking? I think it is very difficult to interpret with the current behavioral setup.

Figure 1, top. Scale bars on the individual fish would be helpful to understand how their size compares to zebrafish. I also suggesting including a zebrafish image for easy comparison. The quality of these images is also somewhat mixed and needs to be more consistent. E.g., the ternetzi and juvenile nigrofasciata seem to be more pixelated than many of the other images.

Figure 6, 'w/ P. tetrazona the 'c' looks to be overlapping the points.

Figure S8 states that the body area was tested against group differences for each measure. There are 12 points per graph, but aren't there 12 x 4 groupings (48)? It's stated in the methods that different test fish were used for each set of 6 naïve fish (so 4 different test fish each). So you should have four different body sizes for each test fish if I understand correctly.

Figure S9, correlations say they are n=4, but there are only 3 points per plot and one has 12 points. Also, correlations with only 3-4 points are not very convincing. Even so, it doesn't appear any statistical tests are performed here; which one's of these correlations are significant. Certainly not all of them.

Major points

What are the sex of the zebrafish? It's well established in the adult zebrafish behavioral field at this point that there are sex differences in behavior like bottom dwelling and distance travelled. Although the effect of sex on shoaling hasn't been studied extensively to my knowledge (but see Ruhl et al, 2009), it is likely that the mix of male and female fish would be expected to affect

shoaling size/density/dynamics. If this wasn't explicitly controlled for (e.g., always using 3 male and 3 female zebrafish for each group) then some of the findings may reflect different sex ratios instead of differences in responses to 'test' fish.

The control group used for presentation in the figures and statistical comparison is inappropriate. The correct way to do the comparisons would be to use controls for fish in each group exposed to a specific predator, not pooling all 288 fish for comparison. Using a pool of all 288 for the control and doing comparisons between all these fish and each individual group overly inflates the statistical power. Indeed, this is one way to mitigate variation in the composition of each group, albeit imperfectly.

Treating each individual fish as an independent biological replicate in several of the analyses (e.g., figures 2 and 3) is inappropriate. This is because each collection of six fish are together at once with the test fish. The unit of manipulation is one test fish and six zebrafish. Because the six zebrafish interact with each other, they are not independent replicates. Independence of replicates is a key assumption of the statistical test used here (Krusal-Wallis) which is violated by the experimental design. A mixed-effects model would be needed here.

The experimental design where 'test' fish are put into the same tank as zebrafish makes the data extremely difficult to interpret. This is because the behavior of the test fish will invariably affect the behavior of the zebrafish as they interact. So fish spending more time at the top of the tank (which would normally be interpreted as reduced anxiety-like behavior or increased risk taking) would not make sense if the test fish was swimming vigorously around the bottom of the tank. This issue is compounded by the fact that there is likely variation in the behavior of the test fish. A more controlled way to have performed these studies would be to have exposed zebrafish to test fish through a transparent barrier (perhaps with perforations if chemical cues are deemed important). As it stands, I do not find the experimental design convincing.

References

Ruhl, N., McRobert, S. P., & Currie, W. J. (2009). Shoaling preferences and the effects of sex ratio on spawning and aggression in small laboratory populations of zebrafish (*Danio rerio*). *Lab animal*, 38(8), 264-269.

Reviewer 2

Comments for the author

This manuscript by Kurnia and colleagues presents a comparative analysis of zebrafish fear responses to eight different fish species using 3D locomotion tracking assays. The 3D tracking methodology represents a clear advancement over 2D analysis, demonstrated by the finding that only 2 of 9 endpoints show strong correlation between viewing dimensions. The comprehensive comparison across multiple fish species provides valuable data for behavioural researchers. The core experimental approach is sound and the datasets comprehensive, however a few methodological and analytical concerns require attention but with improvements to figure accessibility, legend descriptions and discussion of the biological significance of the findings, this manuscript would make a solid contribution to zebrafish behavioural research.

Major comments:

- The experimental design appropriately includes tiger barb as a negative control and convict cichlid as a positive control. However, additional controls would strengthen the interpretation of fear responses. The authors should consider including an empty tank condition to establish baseline behaviour and an inanimate object of similar size to distinguish fear responses from general responses to novel stimuli.
- The statistical analysis appears adequate with Kruskal-Wallis tests and Dunn's multiple comparisons but some details require clarification. The manuscript states sample sizes (e.g. n=24 per treatment group, n=288 control) but it would be helpful to clarify whether this represents

individual fish or group measurements, particularly for the behavioural endpoints that may involve group-level calculations.

- The correlation analysis between fish body size and behavioural responses reveals no correlation across different species but shows correlation within convict cichlids of different ages. This suggests species-specific factors beyond size drive responses, which deserves more discussion.
- The colour preference findings for *G. ternetzi* variants are interesting but the manuscript could benefit from more quantitative analysis of approach/avoidance behaviours rather than relying primarily on clustering analysis.

Minor comments:

- Figure organisation needs improvement. Some figures are referenced in the Methods section (e.g. page 4, line 139 mentions "Figure 1") when figures should be introduced in the Results section where data are presented and discussed.
- Figure 6 requires a more descriptive legend explaining what the letter groupings (a, b, c, etc.) represent for statistical significance, as readers cannot interpret the data without this information.
- Figure 7 swimming pattern visualisations would benefit from scale indicators and clearer tank boundary markings to help readers assess spatial relationships between zebrafish and test fish.

Reviewer's Responses to Questions

Experimental quality

Does each figure have the proper controls?

If 'No', please indicate reasons in Comments for Author box below.

Reviewer #1:

- No

Reviewer #2:

- No

Were the data analyzed using appropriate statistical tests?

If 'No', please indicate reasons in Comments for Author box below.

Reviewer #1:

- No

Reviewer #2:

- Yes

Reproducibility

Were experiments performed using adequate number of biological replicates?

If 'No', please indicate reasons in Comments for Author box below.

Reviewer #1:

- No

Reviewer #2:

- Yes

Does the methods section provide sufficient detail to permit reproducibility?
If 'No', please indicate reasons in Comments for Author box below.

Reviewer #1:

- No

Reviewer #2:

- Yes

Completeness

Are the manuscript's conclusions supported by the data?
If 'No', please indicate reasons in Comments for Author box below.

Reviewer #1:

- No

Reviewer #2:

- Yes

Scholarship

Do the authors cite and discuss the merits of data that would argue for and against their conclusion?
If 'No', please indicate reasons in Comments for Author box below.

Reviewer #1:

- No

Reviewer #2:

- Yes

Does the manuscript title & abstract accurately reflect the contents of the manuscript, without hyperbole?
If 'No', please indicate reasons in Comments for Author box below.

Reviewer #1:

- Yes

Reviewer #2:

- Yes

First revision

Author response to reviewers' comments

Reviewer 1: Kurnia and colleagues describe a series of experiments where they attempt to determine how groups of zebrafish respond to different fish species. While an interesting idea, there are some significant flaws in the experimental design that dampen my enthusiasm. There is also some sloppiness in the figures and captions and the overall writing also needs to be improved.

Minor points

Line 39, Fear is not a 'sensor', but a response to perceived threat.

We thank the reviewer for the correction, as indeed, fear is a response when an individual perceives threats; that is, putting a 'sensor' in this context would be unsuitable. Therefore, the used term in the manuscript was adjusted accordingly (line 44).

Line 51, this is awkwardly written, zebrafish don't elicit responses to anxiety. They elicit anxiety like responses.

We agree with the reviewer's comment. The sentence was indeed, was not inappropriately written as intended, which might cause confusion to the readers; thus, the sentence was revised to "... to elicit response to anxiety-like responses, stress response, and fear response ..." (line 56).

Line 54, this is awkwardly written, freezing is not related to zebrafish response to fear. It is a response indicative of fear.

We appreciate the reviewer for pointing out this matter. As the reviewer mentioned, it is true that freezing is not related to the zebrafish response to fear; instead, it is a response indicative of fear when perceiving threats. Therefore, the sentence was revised to "... freezing indicates zebrafish fear response and anxiety-like responses." (line 60).

Line 57, citation?

We thank the reviewer for asking about the reference regarding the previous study on the zebrafish behavioral changes in the presence of *Nandus nandus*. In the updated manuscript, the citation regarding the reference has been added in line 62.

Line 58, 'we would like to observe' is incorrect. "We examined zebrafish interspecies responses..."

Thank you for highlighting the incorrectly written sentence in the manuscript. The mentioned sentence was revised according to the suggestion from the reviewer (line 63).

Lines 59-66. Are these fish natural predators of zebrafish? Does their natural range overlap with those of zebrafish? Why would it be expected that zebrafish would respond fearfully to some fish species versus others?

We appreciated the reviewer for raising these critical questions. We had tried our best to obtain any information on the natural environment of all of the fish species that were tested in the present study, and whether they share a natural environment and have a role as a natural predator to zebrafish from previous studies or the literature. Unfortunately, only two fish species used in the present study, tiger barb (*P. tetrazona*), which were known to live side-by-side with small fishes, including zebrafish and convict cichlid (*A. nigrofasciata*) which has been used to induce fear response in zebrafish had been reported in a previous study. There is minimal information regarding the interaction between zebrafish and the rest of the fish species used in this study, therefore they have not been classified as zebrafish predator. Additionally, the natural range did not overlap

between zebrafish which originated from South Asia and the other fish species used in this study which originated from Southeast Asia, Central America, South America, and South Africa. Based on the literature study on *P. tetrazona* and *A. nigrofasciata*, we suspected that the fear response in zebrafish might be correlated with the body size of the other fish, responding fearfully to some fish with a relatively big body size versus others, considering the relatively bigger size of the adult convict cichlid compared to the adult tiger barb. Thus, although there has been no report on the interaction between the other fish species and zebrafish, we felt that it was intriguing to verify whether zebrafish behavioral response towards other species are related to body size or not, though based on the current results no correlation between zebrafish behavior response to the predator fish's body size was observed (lines 72-84).

Line 179, the word 'palpable' here is inappropriate.

We thank the reviewer for the suggestion and agree with the reviewer regarding the usage of the word "palpable" in the PCA and heatmap clustering analysis section in the materials and methods part. Therefore, the sentence was revised to "PCA and heatmap analysis were used to obtain a more comprehensive visualization ..." (lines 525-529).

Line 566. Should be 'Fish species and their variants used...'

Thank you for the suggestion. We had edited the caption of Figure 1 according to the reviewer's suggestion, as the convict cichlid in different growth stages was used in the present study; thus, '... tested fish species and their variants ...' is more appropriate to be written here to provide better clarity for the readers (line 687).

I would suggest including a table of the different fish species used, their native habitats, and relevant characteristics.

We appreciate the reviewer for the constructive suggestion and agree that the addition of a table that includes detailed information regarding the fish species used and their native habitats and relevant characteristics would be beneficial for the readers to understand more about the fish species that were used in the present study. Therefore, a table that covers fish species information, role in study, Family, known behavior, origin, some additional notes, and references was added to the manuscript as "Table 1. Summary of zebrafish and tested fish behavior." (line 756).

Line 102, units on the size of the holding tank

We thank the reviewer for the reminder. Actually, the written information regarding the size of the holding tank was in cm units. Therefore, we added the units in the manuscript to provide clearer information on the used holding tank (line 466).

Line 109-114: What is the source of other 'test' fish?

Thank you for raising this important question, and we understand the importance of including the source of the tested fish in the current study. Thus, the detailed information regarding the local aquarium vendor where the fish were obtained was added to the manuscript (lines 456-462).

Line 318-319: It is inappropriate to state that zebrafish are relaxed and confident in the presence of tetrazona. Please only describe the behavior without undue anthropomorphism.

We agree with the reviewer that it is inappropriate to state that zebrafish were relaxed and confident in the presence of the tiger barb. Therefore, to avoid anthropomorphism in the manuscript, the sentence was revised to only mention the results from the calculated behavior endpoints (lines 276-278).

Line 321-322: Staying near the top of the tank is not usually interpreted as 'increased confidence'. Increased confidence in what? It has been interpreted as a reduction in anxiety-like behavior or more willingness to take risk. However, I don't know that you can make this interpretation in your data. The issue is that the behavior of the 'test' fish is unaccounted for. If the test fish is swimming near the bottom of the tank and preventing the zebrafish from spending time there, does that make the top dwelling behavior a sign of decreased anxiety or increased risk taking? I think it is very difficult to interpret with the current behavioral setup.

We appreciated the detailed comments from the reviewer regarding the interpretations of the obtained behavior results. After considering the suggestion, we also feel that 'confidence' might be inappropriate to be used to interpret the data in this context. Therefore, the sentence was revised to "..., which means *P. tetrazona* does not induce anxiety-like or predator avoidance response from

zebrafish” to avoid confusion for the readers. This claim was based on the vertical exploration of zebrafish in the three sections of the test tank (top, middle, and bottom), which was found to be unaffected in the presence of *P. tetrazona*, meaning that zebrafish did not stay near the top of the tank throughout the majority of the time. While it is true that the behavior of the ‘test’ fish was unaccounted for, as the reviewer mentioned, we believe that the observed unchanged vertical exploration is quite convincing that the presence of *P. tetrazona* did not majorly change zebrafish exploratory behaviors. Moreover, although we take the test fish's behaviors into account, we are afraid that it would also be difficult to interpret their behaviors, considering the scarcity of literature on these fish's behaviors compared to zebrafish. In addition, this conclusion was also supported by the reduction in thigmotaxis of zebrafish, which, based on the previous study, generally, this phenomenon indicates that there was no anxiety-like and predator avoidance behavior observed in the tested zebrafish (lines 276-281).

Gerlai, R. T. (2020a). *Behavioral and neural genetics of zebrafish*: Academic Press.

Figure 1, top. Scale bars on the individual fish would be helpful to understand how their size compares to zebrafish. I also suggesting including a zebrafish image for easy comparison. The quality of these images is also somewhat mixed and needs to be more consistent. E.g., the ternetzi and juvenile nigrofasciata seem to be more pixelated than many of the other images.

Thank you for pointing out this matter, and we acknowledge the importance of including information regarding the used fishes' body area to help the readers comprehend the findings of the current study. Therefore, the average body area of each tested fish relative to the average zebrafish body area based on the data that are displayed in Figures S7, S8, Table S2, and average zebrafish body area was added to Figure 1 since we believe that this type of information is easier to understand by the readers while still enabling the readers to obtain information on the approximate body area of every tested fish. Next, in terms of the image quality of the several tested fishes, which was not consistent with the others as the reviewer mentioned, we have resolved this issue by updating the inconsistent quality images with higher resolution in Figure 1.

Figure 6, 'w/ *P. tetrazona* the 'c' looks to be overlapping the points.

We appreciate the detailed comment from the reviewer. We had confirmed the mistake and revised the figure to avoid overlapping letters with the points as the reviewer suggested (Figure 6).

Figure S8 states that the body area was tested against group differences for each measure. There are 12 points per graph, but aren't there 12 x 4 groupings (48)? It's stated in the methods that different test fish were used for each set of 6 naïve fish (so 4 different test fish each). So you should have four different body sizes for each test fish if I understand correctly.

We thank the reviewer for the detailed question. Actually, in Figure S7 (previously Figure S8), it is true that there are a total of 48 tested fish and their variants, as the reviewer mentioned. However, due to the different concept of a data presentation applied in the previous version of the manuscript, we used the average body size of each tested fish; thus, there were only 12 points per graph. Therefore, we edited the figure to have a total of 48 points in response to the reviewer's comment. In addition, we also used the data for 6 zebrafish per tank to show the individual variability compared to the previous average data. These data were used due to high variability in behavioral data; as such, the use of average data per tank might not appropriately represent the behavior of individual zebrafish.

Figure S9, correlations say they are n=4, but there are only 3 points per plot and one has 12 points. Also, correlations with only 3-4 points are not very convincing. Even so, it doesn't appear any statistical tests are performed here; which one's of these correlations are significant. Certainly not all of them.

Thank you for the detailed comments from the reviewer. Similar to the point above, we revised Figure S8 (previously Figure S9) so now it contains 12 points in each group rather than using the average value for body area as shown in the previous version of the manuscript. Similar to Figure S7, we also used the data for 6 zebrafish per tank to show the individual variability compared to the previous average data, in the updated revision.

Major points

What are the sex of the zebrafish? It's well established in the adult zebrafish behavioral field at this point that there are sex differences in behavior like bottom dwelling and distance travelled.

Although the effect of sex on shoaling hasn't been studied extensively to my knowledge (but see Ruhl et al, 2009), it is likely that the mix of male and female fish would be expected to affect shoaling size/density/dynamics. If this wasn't explicitly controlled for (e.g., always using 3 male and 3 female zebrafish for each group) then some of the findings may reflect different sex ratios instead of differences in responses to 'test' fish.

We appreciated the detailed questions from the reviewer regarding this matter. Actually, in the present study, we used adult zebrafish of mixed gender with a randomly sampled male/female ratio, considering the focus of the study to evaluate the effect of the presence of various fishes on zebrafish behaviors in general. While we are fully aware that gender might have some effect on the shoaling behaviors in zebrafish, as mentioned by the reviewer, we believe that it would be more suitable to be assessed in future studies to maintain the focus of the present study. In this consideration, we also added this matter as one of the directions that could be applied in future studies in the discussion section to let the readers acknowledge the possibilities of the gender effects in zebrafish shoaling behaviors (lines 406-410).

The control group used for presentation in the figures and statistical comparison is inappropriate. The correct way to do the comparisons would be to use controls for fish in each group exposed to a specific predator, not pooling all 288 fish for comparison. Using a pool of all 288 for the control and doing comparisons between all these fish and each individual group overly inflates the statistical power. Indeed, this is one way to mitigate variation in the composition of each group, albeit imperfectly.

We thank the reviewer for the constructive suggestion regarding the statistical comparison in the figures and agree that it might not be the most appropriate way to do the comparison since, indeed, in this way, there was a high possibility of inflated statistical power. Therefore, we changed the data so now they are not pooling all 288 fish for comparison; instead, we divided the 288 fish into 12 groups according to the tested fish, resulting in 24 zebrafish per treatment (a total of four replications with 6 zebrafish per replication) before and after the addition of tested fishes and their variance (Figures 2 -5).

Treating each individual fish as an independent biological replicate in several of the analyses (e.g., figures 2 and 3) is inappropriate. This is because each collection of six fish are together at once with the test fish. The unit of manipulation is one test fish and six zebrafish. Because the six zebrafish interact with each other, they are not independent replicates. Independence of replicates is a key assumption of the statistical test used here (Kruskal-Wallis) which is violated by the experimental design. A mixed-effects model would be needed here.

Thank you for pointing out this matter, and we understand the reviewer's point of view regarding the inappropriateness of treating each individual fish as an independent biological replicate. However, this decision was taken to avoid mitigation in the variation of the composition of each group, as also mentioned in the point above, since, to the best of our knowledge, the individual variability in a behavioral study is very high in general. While it is true that the six zebrafish may interact with each other, this interaction might also disappear in the presence of the test fish. Therefore, we believe that merging the data into one may not represent the comprehensive behavioral result and potentially neglect some minor behavioral aspects that might be important to the overall results. Moreover, pulling the six zebrafish into a single replicate might also result in the necessity of testing a higher sample size of zebrafish, which might potentially go against the 3Rs principle that is fundamental for behavior studies. In addition, regarding the statistical tests, we agree with the reviewer that the use of Kruskal-Wallis as a statistical test is unsuitable, especially after separating the groups; thus, to accommodate the change, we used Two-way ANOVA mixed-effect analysis as the new statistical test used in this study.

The experimental design where 'test' fish are put into the same tank as zebrafish makes the data extremely difficult to interpret. This is because the behavior of the test fish will invariably affect the behavior of the zebrafish as they interact. So fish spending more time at the top of the tank (which would normally be interpreted as reduced anxiety-like behavior or increased risk taking) would not make sense if the test fish was swimming vigorously around the bottom of the tank. This issue is compounded by the fact that there is likely variation in the behavior of the test fish. A more controlled way to have performed these studies would be to have exposed zebrafish to test fish through a transparent barrier (perhaps with perforations if chemical cues are deemed important). As it stands, I do not find the experimental design convincing.

We appreciated the reviewer for the detailed comment regarding this matter. First, it is indeed

that there is a possibility that the behavior of the test fish will invariably affect the behavior of the zebrafish as they interact when they are put into the same tank as zebrafish, as the reviewer mentioned. However, based on our experience, a more controlled way of exposure, such as the use of a transparent barrier, would also face a similar problem, although it may be in a less robust manner, since even though they are separated, the robustness of test fish activity may also affect zebrafish behavior responses. Further, while it is also true that the test fish position would greatly affect the zebrafish position and eventually affecting behavior interpretation of the tested zebrafish, we believe that deducing the overall behavior responses of zebrafish solely based on the well-known their behavior endpoints, such as reduced anxiety when they are in the top portion of the tank, may be not suitable in this case. Therefore, it is the reason why we also calculated the average distance between the tested fish and each zebrafish as one of the most important behavior endpoints, as it showed the physical distance between zebrafish and tested fish, which is related to avoidance behavior. Taken together, by observing the result from zebrafish distance to tested fishes, time spent in the top portion of the tank, and projected 3D trajectories, we are able to observe the avoidance by zebrafish from the tested fishes, by avoiding the vertical space occupied by the tested fishes. In addition, the approach suggested by the reviewer with the use of a transparent barrier has been replicated in the previous study by Ladu *et al.*, (without the perforations) which used *Astronotus ocellatus*, an allopatric zebrafish predator that stay in the bottom of the tank, thus making zebrafish to stay in the top area rather than the bottom, which was similar with the current zebrafish behavior in the presence some of the tested fishes, including *A. nigrofasciata* and *G. ternetzi* and their variants. Furthermore, we also believe that the current design might represent the real-life condition better, albeit with limited space, which can be improved in a later study. However, we understand the necessity of a more controlled environment in this type of study; thus, we also add this issue as a limitation of our study in the discussion section (lines 401-406)

Ladu, F., Bartolini, T., Panitz, S. G., Chiarotti, F., Butail, S., Macrì, S. & Porfiri, M. (2015). Live predators, robots, and computer-animated images elicit differential avoidance responses in zebrafish. *Zebrafish* 12, 205-214.

References

Ruhl, N., McRobert, S. P., & Currie, W. J. (2009). Shoaling preferences and the effects of sex ratio on spawning and aggression in small laboratory populations of zebrafish (*Danio rerio*). *Lab animal*, 38(8), 264-269.

Reviewer 2: This manuscript by Kurnia and colleagues presents a comparative analysis of zebrafish fear responses to eight different fish species using 3D locomotion tracking assays. The 3D tracking methodology represents a clear advancement over 2D analysis, demonstrated by the finding that only 2 of 9 endpoints show strong correlation between viewing dimensions. The comprehensive comparison across multiple fish species provides valuable data for behavioural researchers. The core experimental approach is sound and the datasets comprehensive, however a few methodological and analytical concerns require attention but with improvements to figure accessibility, legend descriptions and discussion of the biological significance of the findings, this manuscript would make a solid contribution to zebrafish behavioural research.

Major comments:

- The experimental design appropriately includes tiger barb as a negative control and convict cichlid as a positive control. However, additional controls would strengthen the interpretation of fear responses. The authors should consider including an empty tank condition to establish baseline behaviour and an inanimate object of similar size to distinguish fear responses from general responses to novel stimuli.

We appreciate the reviewer for the constructive suggestions and acknowledge the necessity of the additional controls to strengthen the interpretation of fear response. Actually, the zebrafish behaviors prior to the presence of the tested fish (an empty tank condition) were also recorded in the current study as the control, as the reviewer mentioned. However, in the previous version of the manuscript, we reported this data as a single group that consisted of 288 zebrafish. However, in the updated version of the manuscript, we decided to divide this control group into 12 groups matching their tested fish counterpart according to the reviewers' suggestion to provide a more appropriate control for the analysis. We hope that this control is sufficient enough to strengthen the interpretation of fear responses since although we agree that it is possible to also induce zebrafish fear response through inanimate objects such as a robot based on zebrafish allopatric

predator as the reviewer mentioned, in the current situation, we have difficulties in providing this inanimate moving object that was used in previous studies and unfortunately, according to Ladu *et al.*, an immobile object is seems not enough to stimulate zebrafish fish response. Thus, we think that it is intriguing to conduct such an experiment using inanimate objects in the future as an additional control, and this future direction was added to the discussion section (lines 400-401).

Ladu, F., Bartolini, T., Panitz, S. G., Chiarotti, F., Butail, S., Macrì, S. & Porfiri, M. (2015). Live predators, robots, and computer-animated images elicit differential avoidance responses in zebrafish. *Zebrafish* 12, 205-214.

- The statistical analysis appears adequate with Kruskal-Wallis tests and Dunn's multiple comparisons but some details require clarification. The manuscript states sample sizes (e.g. n=24 per treatment group, n=288 control) but it would be helpful to clarify whether this represents individual fish or group measurements, particularly for the behavioural endpoints that may involve group-level calculations.

Thank you for pointing out this matter, as we also realize that some details regarding the sample size in the present study require clarification. As mentioned above and also detailed in the manuscript, each tested fish was tested in four biological replicates, with each replicate consisting of six zebrafish; thus, a total of 24 zebrafish were tested for every tested fish. Furthermore, regarding the used statistical tests, two-way ANOVA with mixed effects analysis was used in the updated manuscript to accommodate the new comparison (before and after the addition of tested fishes), as the data arrangement was changed as mentioned above. We hope that this explanation is clear enough for the reviewer to understand all the information regarding the sample size and statistical tests that are used in the present study.

- The correlation analysis between fish body size and behavioural responses reveals no correlation across different species but shows correlation within convict cichlids of different ages. This suggests species-specific factors beyond size drive responses, which deserves more discussion.

We thank the reviewer for the comments. We agree that, based on the current findings, especially the results regarding convict cichlids, it suggests species-specific factors beyond size drive responses. In order to support this finding, we added important findings from the previous study that showed a relatively similar phenomenon to the current study. In their study, they tested the predator avoidance behavior of *Poecilia Mexicana* by exposing them to four different fish species. Based on their results, it was found that the most robust predator avoidance behaviors was observed during the presence of the fish that was more active than the other fishes, highlighting the possibility of the common morphological features as the major factor in affecting fish predator avoidance behavior rather than the body size, which might also help in elucidating the similar phenomenon observed in the present study.

Bierbach, D., Schulte, M., Herrmann, N., Zimmer, C., Arias-Rodriguez, L., Indy, J. R., Riesch, R. & Plath, M. (2013). Predator avoidance in extremophile fish. *Life* 3, 161-180.

- The colour preference findings for *G. ternetzi* variants are interesting but the manuscript could benefit from more quantitative analysis of approach/avoidance behaviours rather than relying primarily on clustering analysis.

We appreciate the constructive suggestions from the reviewer and agree that discussing the color preference findings based on the quantitative analysis from the behavior endpoints calculation would be better than primarily relying on the clustering analysis alone. Therefore, we added a new discussion section regarding this matter to highlight the significant behavior differences between every color variant of *G. ternetzi* to each other in relative to their respective control group. The observed behavior differences include the green *G. ternetzi* that showed a significantly longer time in the top stay compared to the other colors. Furthermore, the average zebrafish-tested fish distance was also statistically different between green to red and orange *G. ternetzi*, with zebrafish tending to stay farther from green *G. ternetzi* compared to red and orange *G. ternetzi*. Finally, the tested zebrafish formed a tight shoaling formation in the presence of green *G. ternetzi*, while shoal volume was unchanged during the presence of red and orange *G. ternetzi*. We believe that this newly added explanation of the obtained quantitative results in the present study in the discussion section is sufficient to highlight the color preference findings for *G. ternetzi* variants (lines 296-304).

Minor comments:

- Figure organisation needs improvement. Some figures are referenced in the Methods section (e.g.

page 4, line 139 mentions "Figure 1") when figures should be introduced in the Results section where data are presented and discussed.

We thank the reviewer for the suggestion regarding the figure organization in the manuscript. It is true that some figures may be less suitable to be included in some sections of the manuscript. Therefore, these figures were relocated accordingly. As an example, Figure 1, which contains a graphical study design summary that includes the brief information on the fishes that were tested in this study to induce fear response on zebrafish and the applied behavior assays (recording setup, recording time, software used, etc.), was moved to the introduction section as it covers the study outline of the present study (lines 108-109).

- Figure 6 requires a more descriptive legend explaining what the letter groupings (a, b, c, etc.) represent for statistical significance, as readers cannot interpret the data without this information. Thank you for the reminder regarding the letters that signify the statistical differences in Figure 6. Additional information regarding the letter was added to the figure's caption. In the figure, different letter signifies statistical differences between groups with a *P* value that was less than 0.05, which were based on the conducted Kruskal-Wallis with Dunn's multiple comparison tests on the data. For example, *P. tetrazona* is labelled as "ac", which means statistically, the values of the behavior endpoint in this group were not statistically different compared to *K. biccirhis*, which is labelled "a", or *C. macracanthus*, which is labeled "cd" since both of these fishes share the same letter as *P. tetrazona*. However, if compared to Adult *A. nigrofasciata* (b), statistically significant differences were observed. We also have added the statistical significance information in the figure captions within the manuscript, as the group with different letter have statistical significance of $p < 0.05$.

- Figure 7 swimming pattern visualisations would benefit from scale indicators and clearer tank boundary markings to help readers assess spatial relationships between zebrafish and test fish.

We thank the reviewer for the constructive suggestion in the visualization of Figure 7. Actually, every boundary that are illustrated in each fish's trajectory already represents a relatively comparable condition to the applied behavior recording setup which used an acrylic test tank with $20 \times 20 \times 20$ cm of size and since only ~6 liters of water was used, the height of the water was only ~15 cm deep as shown in the each axis of the trajectories' boundary in the figure. In addition, we also make the boundary lines in the figure bolder and in a different color to help readers easily distinguish the fish trajectories and the boundaries. Finally, to further aid the readers in comprehending the figure, we have added more detail to the figure's caption, which in the updated version becomes "Figure 7. Visualization of six representative zebrafish (black) and a tested fish (red) swimming patterns during a shared environment test (10 minutes) in a 3D tank, with water filled to 75% of maximum volume (final dimension of 20 cm (X-axis) \times 20 cm (Y-axis) \times 15 cm (Z-axis))."

Second decision letter

MS ID#: bio.062110R1

MS Title: Comparative analysis of Zebrafish fear responses to eight different fish species using 3D locomotion tracking assays

Authors: Kevin Adi Kurnia, Gilbert Audira, Michael Edbert Suryanto, Tzong-Rong Ger and Chung-Der Hsiao

I am happy to tell you that your manuscript has been accepted for publication in Biology Open, pending our standard publication integrity checks. It was accepted on 11th September 2025.